# COGNOS: Universal Enhancement for Time Series Anomaly Detection via Constrained Gaussian-Noise Optimization and Smoothing

Wenlong Shang [1]   Shihao Tian [1]   Xutong Wan [1]   Peng Chang [2]

## Abstract

Reconstruction-based methods are a dominant paradigm in time series anomaly detection (TSAD), however, their near-universal reliance on Mean Squared Error (MSE) loss results in statistically flawed reconstruction residuals. This fundamental weakness leads to noisy, unstable anomaly scores, hindering reliable detection. To address this, we propose Constrained Gaussian-Noise Optimization and Smoothing (COGNOS), a universal, model-agnostic enhancement framework that tackles this issue at its source. COGNOS introduces a novel Gaussian-White Noise Regularization strategy during training, which directly constrains the model's output residuals to conform to a Gaussian white noise distribution. This engineered statistical property creates the ideal precondition for our second contribution: Adaptive Residual Kalman Smoother that operates as a statistically robust estimator to denoise the raw anomaly scores. Extensive experiments on multiple benchmarks demonstrate that COGNOS consistently and substantially enhances the performance of state-of-the-art backbones, validating the efficacy of coupling statistical regularization with adaptive filtering.

## 1. Introduction

Time Series Anomaly Detection (TSAD) is a critical task in reliable machine intelligence, essential for monitoring complex systems in industrial, aerospace, and IT domains (Jia

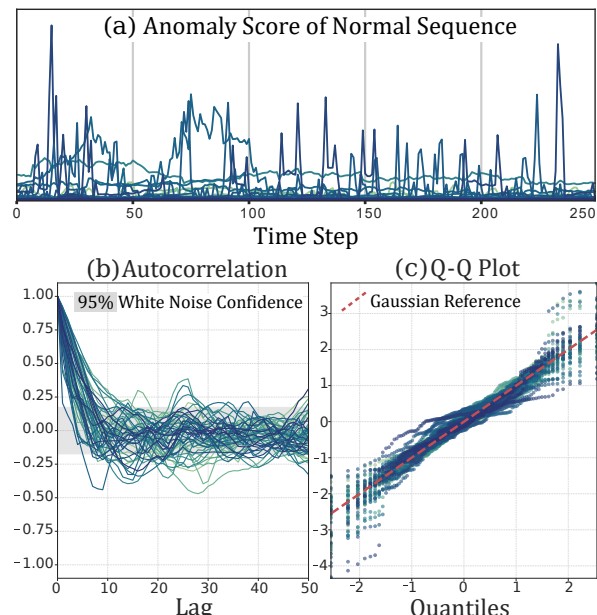

*Figure 1.* We analyze the TimesNet model on the SWaT dataset: (a) The resulting anomaly scores are highly noisy, leading to a high noise floor during normal operation which makes actual anomalies difficult to detect. (b) The autocorrelation plot shows significant temporal correlations persist in the residuals, suggesting that the model failed to capture all predictable patterns. (c) The Q-Q plot reveals that reconstruction residuals are strongly non-Gaussian, indicating that the underlying noise was poorly modeled.

et al., 2025). The prevailing paradigm leverages Deep Neural Networks (DNNs), ranging from foundational Autoencoders (AE) (Hinton & Salakhutdinov, 2006; Sakurada & Yairi, 2014; Chevrot et al., 2022) and Recurrent Neural Networks (RNN) (Hochreiter & Schmidhuber, 1997; Hundman et al., 2018; Liu et al., 2022a) to modern Transformers (Vaswani et al., 2017; Xu et al., 2022; Kang & Kang, 2024; Bai et al., 2025), have been employed to learn complex dynamics of normal data. The core premise is intuitively simple: a model trained on normal data will exhibit high reconstruction errors when encountering anomalies.

However, a fundamental theoretical gap persists in this paradigm: the **misalignment between the optimization objective and the inference requirement.** Standard approaches predominantly optimize the Mean Squared Error (MSE), which corresponds to Maximum Likelihood Estima-

[1]School of Information Science and Technology, Beijing University of Technology, China [2]Beijing Key Laboratory of Computational Intelligence and Intelligent System, Beijing University of Technology, China. Correspondence to: Peng Chang <changpeng@emails.bjut.edu.cn>.

*Proceedings of the 43rd International Conference on Machine Learning*, Seoul, South Korea. PMLR 306, 2026. Copyright 2026 by the author(s).

The code is available at: https://github.com/giansha/COGNOS

tion (MLE) only under the strict assumption that residuals are independent and identically distributed (i.i.d.) Gaussian noise. In practice, as illustrated in Fig. 1, deep models prioritize minimizing global energy but fail to ensure the statistical purity of the residuals. The resulting residuals often exhibit **non-Gaussianity** and **Temporal correlation**.

This gap creates a dilemma: we possess powerful feature extractors, yet we employ a statistically flawed metric for the final decision-making. To bridge this gap, we propose a shift from purely architectural improvements to residual engineering. We argue that for an anomaly detector to be robust, it must satisfy the **Wold Decomposition Theorem**, strictly separating deterministic dynamics from stochastic white noise.

To this end, we introduce **COGNOS** (**CO**nstrained **G**aussian-**N**oise **O**ptimization and **S**moothing), a universal framework designed to enforce these statistical preconditions explicitly. First, we propose the **Gaussian-White-Noise Regularized (GWNR) Loss**, which acts as a soft constraint during training. It forces the backbone to push all deterministic information into the latent space, ensuring the output residuals approximate Gaussian white noise. Second, leveraging this engineered property, we introduce the **Adaptive Residual Kalman Smoother (ARKS)**. It employs a hypothesis-driven "Circuit Breaker" mechanism that dynamically switches between noise suppression (smoothing) and zero-lag tracking (detection) based on the statistical significance of the innovation.

Our approach offers a structural rather than incremental improvement. While recent works focus on designing increasingly complex backbones, we show that correcting the statistical misalignment of the residuals yields substantially larger performance gains than architectural refinements alone. Our contributions are as follows:

- We identify the mismatch between the MSE objective and the statistical requirements of anomaly scoring and propose the GWNR loss to explicitly shape reconstruction residuals into Gaussian white noise.

- We design a ARKS module that acts in synergy with our regularization, leveraging the engineered residual properties to achieve optimal denoising of anomaly scores and improve stability.

- Extensive experiments on multiple benchmarks demonstrate that our framework consistently enhances the performance of various models without requiring expensive hyperparameter tuning of the backbone, validating the efficacy of aligning residual physics with statistical estimation.

## 2. Related Work

### 2.1. Evolution of Reconstruction-based Architectures

Reconstruction-based methods constitute the mainstream of unsupervised anomaly detection. The fundamental premise is that models trained on normal data will yield high reconstruction errors for anomalous patterns. Early works employed **Autoencoders (AE)** (Sakurada & Yairi, 2014; Chevrot et al., 2022) and **LSTM-based** (Malhotra et al., 2015; Hundman et al., 2018; Liu et al., 2022a) networks, such as LSTM-AE (Kieu et al., 2018) combined recurrence with encoder-decoder structures to better model complex temporal dynamics. While effective for short-term dependencies, these recurrent architectures often struggle with long-term correlations and parallelization efficiency. To address the limitations of RNNs, **Transformer-based** architectures introduced self-attention mechanisms to model global dependencies. The Anomaly Transformer (Xu et al., 2022) explicitly maximizes the discrepancy between prior and series associations to highlight anomalies. Other variants further adapted the attention mechanism to focus on distinct anomaly patterns (Kang & Kang, 2024; Bai et al., 2025). The current state-of-the-art has shifted towards **decomposing complex signals into interpretable components**. TimesNet (Wu et al., 2023) transforms 1D series into 2D tensors to capture intra-period variations, while ModernTCN (Luo & Wang, 2024) revisits convolutional structures with large receptive fields. Most recently, methods like TimeMixer++ (Wang et al., 2025) and CrossAD (Li et al., 2025) explicitly leverage multi-scale mixing and cross-dimension dependencies to handle complex industrial dynamics. Similarly, KAN-AD (Zhou et al., 2025) and CATCH (Wu et al., 2025) utilizes Kolmogorov-Arnold Networks or patching technics for efficient frequency signature capture. Despite their architectural sophistication, these models predominantly optimize the Mean Squared Error (MSE). Our work complements these backbones; rather than proposing a new architecture, we introduce a universal framework that addresses the statistical deficiencies of the MSE objective itself.

### 2.2. Representation vs. Reconstruction Paradigms

Parallel to reconstruction, contrastive learning approaches focuses on learning **discriminative latent representations**. Universal methods like TS2Vec (Yue et al., 2022), SimMTM (Dong et al., 2023) optimize the distance between positive and negative pairs in the latent space, and anomaly detection methods like DCdetector (Yang et al., 2023), CARLA (Darban et al., 2025a) further refine this paradigm by integrating dual-attention mechanisms or anomaly injection to enhance distinctiveness. However, such methods lack physically interpretable residuals. COGNOS enhances its reliability by enforcing statistical rigor on the output

residuals, a dimension often overlooked by representation learning literature.

## 2.3. Statistical and Signal Processing Techniques in TSAD

Researchers have also explored integrating signal processing priors to enhance robustness. **Wavelet Transforms** have been used to decompose series for multi-resolution analysis (Mallat, 1989; Yao et al., 2023), while **Maximum Mean Discrepancy (MMD)** (Gretton et al., 2012) has been employed to align distribution shifts (Darban et al., 2025b). A parallel direction involves explicit filtering, such as coupling deep networks with **Kalman Filters** (Kalman, 1960) to create hybrid architectures (Huang et al., 2023; Ma et al., 2024). In contrast, COGNOS strictly decouple the process. By enforcing Gaussian white noise properties during training, we establish the necessary conditions for our Adaptive Residual Kalman Smoother (ARKS) to operate optimally.

## 3. Problem Formulation and Inductive Bias

Let $\mathcal{X} = \{\mathbf{X}^{(i)}\}_{i=1}^{N}$ denote a dataset of multivariate time series, where each sample $\mathbf{X} \in \mathbb{R}^{T \times F}$ consists of $T$ timestamps and $F$ channels. We posit that any observed series $\mathbf{X}_t$ is generated by a composite process:

$$\mathbf{X}_t = \underbrace{f_\theta(\mathbf{X}_{<t})}_{\text{Deterministic Trend}} + \underbrace{\mathbf{n}_t}_{\text{Stochastic Noise}} + \underbrace{\mathbf{s}_t}_{\text{Structural Anomaly}}, \quad (1)$$

where $f_\theta$ is a deterministic function capturing predictable dynamics, $\mathbf{n}_t$ represents irreducible background noise, and $\mathbf{s}_t$ is the sparse structural deviation we aim to detect. In normal data, $\mathbf{s}_t = \mathbf{0}$, which corresponds to the standard **Wold Decomposition** form.

In the reconstruction-based paradigm, a deep neural network approximates $f_\theta$ to produce $\hat{\mathbf{X}}_t$, yielding the residual $\mathbf{R}_t = \mathbf{X}_t - \hat{\mathbf{X}}_t$. Ideally, if the model is perfect ($f_\theta \rightarrow$ True Dynamics), the residual should converge to the stochastic noise: $\mathbf{R}_t \rightarrow \mathbf{n}_t$.

To train such reconstruction models, the standard MSE reconstruction objective implicitly acts as Maximum Likelihood Estimation under the assumption that residuals are isotropic Gaussian noise ($\mathbf{R} \sim \mathcal{N}(0, \sigma^2 \mathbf{I})$)[1]. However, in complex industrial environments, simple reconstruction models often fail to capture all deterministic dynamics, leading to residuals that exhibit significant temporal autocorrelation and non-Gaussian distributions.

To enable optimal detection, we must enforce a stricter inductive bias: *The residual of a normal sequence must constitute a valid Null Hypothesis for statistical testing.* Specifically, we enforce that the recovered noise term $\mathbf{n}_t$

(i.e., the residual $\mathbf{R}_t$ under normal conditions) must approximate **Gaussian White Noise**. We formalize this as a multi-objective optimization problem subject to three physical constraints:

**Minimal Energy:** The magnitude $\|\mathbf{R}\|_F$ should be minimized, ensuring maximal information extraction.

**Spectral Flatness:** The Power Spectral Density (PSD) of $\mathbf{R}$ should be uniform, implying temporal independence.

**Gaussianity:** The distribution of residuals should align with the Gaussian family to satisfy the optimality assumptions of Linear Gaussian State Space Models (LG-SSM). This is the necessary condition for the downstream Kalman Filter (ARKS) to be the Minimum Mean Square Error (MMSE) estimator[2].

By enforcing these constraints via GWNR, we engineer the residuals such that any significant deviation in the subsequent inference stage can be rigorously attributed to the structural anomaly term $\mathbf{s}_t$, rather than modeled noise or leakage.

## 4. Methodology

The COGNOS framework consists of two coupled stages: a training stage governed by the **Gaussian-White-Noise Regularized (GWNR) Loss**, and an inference stage enhanced by the **Adaptive Residual Kalman Smoother (ARKS)**. In order to satisfy the weak stationarity assumption of the Wold Decomposition theorem for sequences, we uniformly apply **Normalization** and **Denormalization** methods to the model input and output to counteract non-stationarity and distribution shifts, thereby achieving more stable reconstruction effects (Kim et al., 2022; Liu et al., 2022b). As illustrated in Fig. 2.

### 4.1. Gaussian-White-Noise Regularized (GWNR) Loss

To enforce the inductive bias formulated above, we introduce the GWNR Loss, which regulates the residuals in the time, frequency, and statistical domains.

#### 4.1.1. ACTIVITY-AWARE DYNAMIC MASKING

Industrial time series frequently contain "frozen" periods (constant values) caused by sensor inactivity. While the reconstruction objective must cover these regions to prevent signal drift, enforcing statistical regularization on constant signals is mathematically ill-posed and may introduce numerical instability.

To address this, we introduce a dynamic binary mask $\mathbf{M} \in \{0,1\}^{B \times T \times F}$. Let $\Delta_t = |\mathbf{X}_t - \mathbf{X}_{t-1}|$ be the first-order

---

[1]Analysis detailed in Appendix A.1.

[2]Analysis detailed in Appendix A.2.

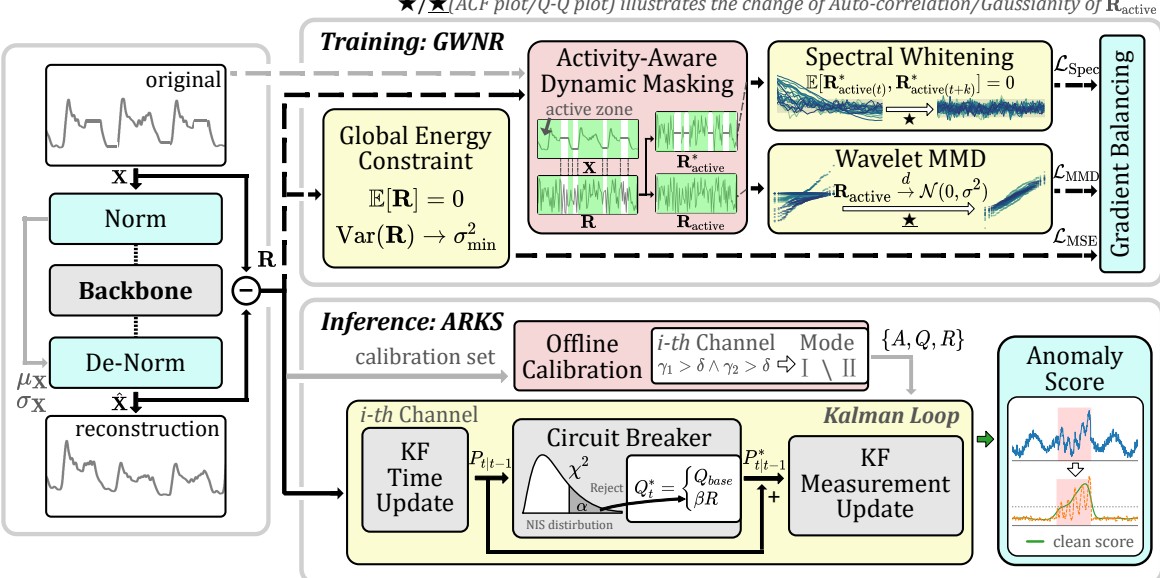

*Figure 2.* COGNOS: Backbones are trained with GWNR Loss, and ARKS is used during inference to generate stable anomaly scores

difference. We employ a morphological dilation operator to generate a mask that covers active regions and preserves transition boundaries:

$$\mathbf{M}_t = \max_{k \in [-w, w]} \mathbb{I}\left(\Delta_{t+k} > \delta\right). \qquad (2)$$

Crucially, while the reconstruction loss is applied globally, the auxiliary regularization terms described below are computed solely on the valid subsequences of the active residuals $\mathbf{R}_{\text{active}} = \mathbf{R} \odot \mathbf{M}$.

### 4.1.2. TIME DOMAIN: GLOBAL ENERGY CONSTRAINT

To ensure faithful reconstruction, we employ the Mean Squared Error as the primary anchor. This term minimizes the total energy of the residual, forcing the model to capture the fundamental signal structure:

$$\mathcal{L}_{\text{MSE}} = \frac{1}{N} \|\mathbf{R}\|_F^2, \qquad (3)$$

where $N = B \times T \times F$. Minimizing $\mathcal{L}_{\text{MSE}}$ creates a compact residual space, which is a prerequisite for the subsequent fine-grained regularization.

### 4.1.3. FREQUENCY DOMAIN: SPECTRAL WHITENING

To eliminate temporal correlations, we aim to maximize the entropy of the residual's spectral distribution. We approximate the spectrum of the stochastic component via the Discrete Fourier Transform (DFT) of the active residuals $\mathbf{R}_{\text{active}}^*$, with zero-padding applied to masked values:

$$\mathbf{Z}(f) = \mathcal{F}(\mathbf{R}_{\text{active}}^*), \quad f = 0, \dots, \frac{T}{2}. \qquad (4)$$

The Power Spectral Density (PSD) is given by $S(f) = |\mathbf{Z}(f)|^2$. To decouple the spectral shape from the residual magnitude, we normalize the PSD into a probability mass function $\tilde{S}$, then we define the Spectral Flatness Loss $\mathcal{L}_{\text{Spec}}$ as the Kullback-Leibler (KL) divergence between $\tilde{S}$ and a uniform distribution $U(f) = 1/K$, which is equivalent to maximizing spectral entropy:

$$\mathcal{L}_{\text{Spec}} = \sum_f \tilde{S}(f) \log \tilde{S}(f) + \log K, \qquad (5)$$

where $K$ is the number of frequency bins. Minimizing $\mathcal{L}_{\text{Spec}}$ forces the residual energy to be distributed equally across all frequencies.

### 4.1.4. STATISTICAL DOMAIN: WAVELET MMD

While spectral whitening ensures temporal independence, it does not guarantee that the noise is Gaussian, which is a strict requirement for the optimality of the downstream Kalman Filter. To bridge this gap, we constrain the higher-order statistics of the residual using Maximum Mean Discrepancy (MMD).

To prevent low-frequency trend leakage from distorting the noise distribution, we employ the Discrete Wavelet Transform (DWT) with an orthogonal Haar basis. The active residual is decomposed into approximation coefficients $\mathbf{A}$ (low-frequency) and detail coefficients $\mathbf{D}$ (high-frequency):

$$(\mathbf{A}, \mathbf{D}) = \text{DWT}_{\text{Haar}}(\mathbf{R}_{\text{active}}). \qquad (6)$$

Due to the orthogonality $\mathbf{A} \perp \mathbf{D}$, we can strictly penalize $\mathbf{D}$ without affecting the model's ability to fit trends captured in $\mathbf{A}$. Since our goal is to shape the noise distribution

without constraining its scale (allowing for channel-specific heteroscedasticity), we apply **Instance Standardization** along the time dimension to the detail coefficients $\mathbf{D}$, then minimize the distributional distance between $\tilde{\mathbf{D}}$ and a standard Gaussian prior $\mathcal{G} \sim \mathcal{N}(0, \mathbf{I})$ via MMD:

$$
\begin{aligned}
\mathcal{L}_{\text{MMD}} = \ & \mathbb{E}_{\tilde{\mathbf{D}}, \tilde{\mathbf{D}}'}[k(\tilde{\mathbf{D}}, \tilde{\mathbf{D}}')] + \mathbb{E}_{\mathcal{G}, \mathcal{G}'}[k(\mathcal{G}, \mathcal{G}')] \\
& - 2\mathbb{E}_{\tilde{\mathbf{D}}, \mathcal{G}}[k(\tilde{\mathbf{D}}, \mathcal{G})],
\end{aligned}
\tag{7}
$$

where $k(\cdot, \cdot)$ is a multi-scale Radial Basis Function (RBF) kernel. This constraint ensures that the residuals possess the Gaussian shape required by the Kalman filter, while the instance standardization preserves the physical variance intrinsic to each channel.

### 4.1.5. GRADIENT BALANCING

The total objective is $\mathcal{L}_{\text{total}} = \mathcal{L}_{\text{MSE}} + \sum_k \alpha_k \mathcal{L}_k$, where $k \in \{\text{Spec}, \text{MMD}\}$. A critical optimization challenge is the magnitude disparity between the reconstruction gradient and the auxiliary gradients, which can lead to instability.

Instead of introducing additional hyperparameters, we employ a dynamic gradient rescaling strategy. We align the norm of auxiliary gradients to the norm of the primary reconstruction task at each step. Let $G_{\text{main}} = \|\nabla_\theta \mathcal{L}_{\text{MSE}}\|_F$ and $G_k = \|\nabla_\theta \mathcal{L}_k\|_F$. The adaptive coefficient $\alpha_k$ is computed as:

$$
\alpha_k = \text{sg}\left( \frac{G_{\text{main}}}{G_k + \delta} \right) \cdot \mathbb{I}(\mathcal{L}_k > \delta),
\tag{8}
$$

where $\text{sg}(\cdot)$ is the stop-gradient operator. The indicator function $\mathbb{I}(\cdot)$ prevents rescaling when the auxiliary loss is negligible (e.g., in purely frozen batches). The final parameter update is driven by:

$$
\nabla_\theta \mathcal{L}_{\text{total}} = \nabla_\theta \mathcal{L}_{\text{MSE}} + \sum_k \alpha_k \nabla_\theta \mathcal{L}_k.
\tag{9}
$$

This mechanism ensures that the "Gaussian whitening" pressure shapes the residuals without overpowering the fundamental reconstruction fidelity.

### 4.2. Adaptive Residual Kalman Smoother (ARKS)

While GWNR ensures that the global residual distribution approximates Gaussian white noise, local anomalies manifest as non-stationary structural breaks that violate the white-noise assumption. To optimally detect these structural deviations, we propose the Adaptive Residual Kalman Smoother (ARKS).

Unlike standard hybrid approaches that couple filtering blindly, ARKS is grounded in the Separation Principle: the backbone removes deterministic dynamics, leaving the filter to act solely as a statistical gatekeeper between stochastic noise $\mathbf{n}_t$ and structural anomalies $\mathbf{s}_t$. We formulate this

as a Linear Gaussian State Space Model (LG-SSM) with a hypothesis-driven adaptive gain.

For a specific channel, the residual process $\{y_t\}$ is modeled as:

$$
\begin{cases}
x_t = Ax_{t-1} + w_t, & w_t \sim \mathcal{N}(0, Q) \\
y_t = Hx_t + v_t, & v_t \sim \mathcal{N}(0, R)
\end{cases}
\tag{10}
$$

where $x_t$ represents the latent anomalous signal. Under normal operation, $x_t \approx 0$ (noise suppression); during anomalies, $x_t$ tracks the deviation.

### 4.2.1. OFFLINE CALIBRATION

To ensure robustness without requiring expensive online parameter learning, ARKS calibrates the system matrices $\{A, Q, R\}$ offline using a calibration set which only contains normal sequences.

We utilize the Method of Moments based on the lag-$k$ autocovariance $\gamma_k$ of calibration residuals to classify each channel into two operating modes:

$$
\text{Mode} \begin{cases}
\textbf{Leakage-Tracking (I)}, \text{ if } \gamma_1 > \delta \wedge \gamma_2 > \delta \\
\textbf{Noise-Suppression (II)}, \text{ otherwise}
\end{cases}.
\tag{11}
$$

**Mode I (Leakage-Tracking):** If the backbone fails to capture complex periodicities, residuals exhibit autocorrelation. We model this as an AR(1) process via Yule-Walker estimation: $A = \gamma_2/\gamma_1$, $Q = \sigma_x(1 - A^2)$, and $R = \gamma_0 - \sigma_x$, where $\sigma_x = \gamma_1^2/\gamma_2$. This configuration forces the filter to track and explain the leakage, preventing it from incorrectly accumulating as an anomaly score.

**Mode II (Noise-Suppression):** If GWNR succeeds, residuals are white. We configure a Random Walk smoother with strong regularization: $A = 1$, $R = \gamma_0$, and $Q = \lambda R$ (with $\lambda \ll 1$). This acts as a low-pass filter ($\hat{x}_t \approx 0$), aggressively suppressing noise to enhance the Signal-to-Noise Ratio (SNR) for true anomalies.

### 4.2.2. RECURSIVE INFERENCE WITH CIRCUIT BREAKER

A standard Kalman Filter with a fixed process noise covariance $Q$ faces a fundamental trade-off: while a small $Q$ is necessary for effective denoising, it introduces significant lag and oversmoothing during abrupt anomalies. To resolve this, we introduce a **Hypothesis-Driven Circuit Breaker Mechanism** that dynamically modulates the epistemic uncertainty of the process. The recursive process at time step $t$ proceeds in three stages:

**1. Prediction (Time Update).** We project the state based on the calibrated dynamics:

$$
\hat{x}_{t|t-1} = A\hat{x}_{t-1|t-1}.
\tag{12}
$$

**2. The Circuit Breaker Mechanism.** At each step $t$, we perform an online Chi-squared test on the Normalized Innovation Squared (NIS),

$$\epsilon_t = \frac{(y_t - \hat{x}_{t|t-1})^2}{P_{t|t-1} + R}. \tag{13}$$

Under the null hypothesis $\mathcal{H}_0$ (residual is Gaussian-White, $\mathbf{s}_t = 0$), $\epsilon_t \sim \chi_1^2$. A value of $\epsilon_t > \tau_\alpha$ indicates a statistically significant *structural break*, i.e., the current observation cannot be explained by the prior dynamics plus observation noise ($\mathbf{s}_t \neq 0$).

To adapt to this structural shift, we trigger the "Circuit Breaker" by inflating the process noise covariance:

$$Q_t^* = \begin{cases} \beta R, & \text{if } \epsilon_t > \tau_\alpha \quad \text{(Anomaly)} \\ Q_{base}, & \text{otherwise} \quad \text{(Smoothing)} \end{cases}, \tag{14}$$

$$P_{t|t-1}^* = A^2 P_{t-1|t-1} + Q_t^*, \tag{15}$$

where $\beta \gg 1$ is a scaling factor, $Q_{base}$ is the base process noise derived from the calibration phase. Mathematically, this inflation forces the Kalman Gain $K_t$ towards 1[3], effectively "short-circuiting" the smoothing dynamics to trust the current observation instantaneously.

**3. Correction (Measurement Update).** We compute the optimal Kalman Gain using the adjusted covariance $P_{t|t-1}^*$ and update the posterior:

$$\begin{cases} K_t = \frac{P_{t|t-1}^*}{P_{t|t-1}^* + R}, \\ \hat{x}_{t|t} = \hat{x}_{t|t-1} + K_t(y_t - \hat{x}_{t|t-1}), \\ P_{t|t} = (1 - K_t)^2 P_{t|t-1}^* + K_t^2 R. \end{cases} \tag{16}$$

### 4.2.3. Anomaly Scoring

The final anomaly score is the energy of the estimated structural signal: $\mathcal{S}_t = \|\hat{x}_{t|t}\|^2$. This score benefits from a double-denoising effect: high-frequency noise is suppressed by the low-pass property of the Kalman smoother ($K_t \ll 1$ under $\mathcal{H}_0$), while trend leakage is absorbed by the process dynamics $A$, isolating only statistically significant anomalies.

## 5. Experiments

### 5.1. Experimental Settings

**Datasets and Evaluation Metrics.** We evaluate our approach on seven widely used real-world time series anomaly detection datasets, namely MSL, SMAP (Hundman et al., 2018), SWaT (Mathur & Tippenhauer, 2016), PSM (Abdulaal et al., 2021), GECCO (Moritz et al., 2018), SWAN (Angryk et al., 2020), and UCR (Wu & Keogh, 2023). These

---

[3]Analysis detailed in Appendix A.3.1.

---

datasets span diverse application domains, such as space exploration, water treatment, service monitoring, etc. Detailed descriptions are provided in Appendix B. To ensure a comprehensive and unbiased evaluation, we utilize a multi-dimensional metric suite that includes both event-based metrics, such as **Standard F1** (Std-F1) (Xu et al., 2018) and **Affiliation-based F1** (Aff-F1) (Huet et al., 2022) and range-based metrics, including **Range-AUC-ROC** (R-A-R), **Range-AUC-PR** (R-A-P), **VUS-ROC** (V-R), and **VUS-PR** (V-P) (Paparrizos et al., 2022).

**Backbones.** To comprehensively evaluate our method, we integrate it with nine backbone models covering diverse architectural paradigms. These include AD models: **CrossAD** (Li et al., 2025), **KAN-AD** (Zhou et al., 2025), **LSTM-AE** (Kieu et al., 2018); universal models: **ModernTCN** (Luo & Wang, 2024), **TimesNet** (Wu et al., 2023), **TimeMixer++** (Wang et al., 2025); and forecasting models: **Autoformer** (Wu et al., 2021), **DLinear** (Zeng et al., 2023), **MICN** (Wang et al., 2023), to demonstrate the broad applicability of our approach. Additional comparisons with the standalone Anomaly Transformer are provided in Appendix D.5.

### 5.2. Experimental Results

**Main Results.** Table 1 and Table 2 present a comprehensive comparison of our framework against the vanilla baselines. The results demonstrate that COGNOS consistently delivers substantial performance gains across all tested architectures, validating its efficacy as a universal enhancement strategy. More details can be found in the Appendix D.1.

*Table 1.* Multi-metric evaluation of anomaly detection performance between the Vanilla method and COGNOS (Ours) across three datasets. Best results are highlighted in **bold**.

| Models | Datasets | MSL | | PSM | | SWAN | |
|---|---|---|---|---|---|---|---|
| | Metircs | Vanilla | Ours | Vanilla | Ours | Vanilla | Ours |
| Autoformer | Std-F1 | 0.7724 | **0.9321** | 0.9098 | **0.9759** | 0.7336 | **0.7950** |
| | Aff-F1 | 0.4008 | **0.9388** | 0.5292 | **0.8743** | 0.1828 | **0.3604** |
| | R-A-R | 0.6942 | **0.7084** | 0.6667 | **0.6954** | 0.9398 | **0.9433** |
| | R-A-P | 0.2366 | **0.2446** | 0.4851 | **0.5235** | 0.9362 | **0.9382** |
| | V-R | 0.6684 | **0.6884** | 0.6349 | **0.6650** | 0.9532 | **0.9542** |
| | V-P | 0.1943 | **0.2039** | 0.4352 | **0.4740** | 0.9074 | **0.9125** |
| KANAD | Std-F1 | 0.8115 | **0.9165** | 0.8946 | **0.9744** | 0.7384 | **0.8049** |
| | Aff-F1 | 0.5360 | **0.9157** | 0.6223 | **0.8628** | 0.2066 | **0.3936** |
| | R-A-R | 0.6646 | **0.6806** | **0.7005** | 0.6799 | 0.9227 | **0.9374** |
| | R-A-P | 0.2030 | **0.2135** | **0.5092** | 0.4740 | 0.9210 | **0.9301** |
| | V-R | 0.6450 | **0.6596** | **0.6846** | 0.6624 | 0.9414 | **0.9460** |
| | V-P | 0.1742 | **0.1799** | **0.4724** | 0.4432 | 0.8950 | **0.8983** |
| TimesNet | Std-F1 | 0.7944 | **0.9160** | 0.9581 | **0.9751** | 0.7371 | **0.8313** |
| | Aff-F1 | 0.4868 | **0.9171** | 0.7828 | **0.8672** | 0.1961 | **0.5249** |
| | R-A-R | **0.6968** | 0.6663 | 0.6742 | **0.7105** | 0.9387 | **0.9413** |
| | R-A-P | **0.2323** | 0.2067 | 0.4932 | **0.5290** | 0.9366 | **0.9394** |
| | V-R | **0.6713** | 0.6500 | 0.6408 | **0.6805** | **0.9544** | 0.9514 |
| | V-P | **0.1925** | 0.1760 | 0.4402 | **0.4787** | 0.9128 | **0.9161** |

The most profound improvements are observed in event-

*Table 2.* Comparison of anomaly detection performance between the Vanilla method and COGNOS (Ours) across seven datasets. The table reports the **Affiliated-F1** metric. Best results are highlighted in **bold**.

| Datasets | GECCO | | MSL | | PSM | | SMAP | | SWAN | | SWaT | | UCR | |
|---|---|---|---|---|---|---|---|---|---|---|---|---|---|---|
| Models | Vanilla | Ours | Vanilla | Ours | Vanilla | Ours | Vanilla | Ours | Vanilla | Ours | Vanilla | Ours | Vanilla | Ours |
| Autoformer | 0.3929 | **0.9426** | 0.4008 | **0.9388** | 0.5292 | **0.8743** | 0.6266 | **0.8521** | 0.1828 | **0.3604** | 0.3967 | **0.9571** | 0.4702 | **0.7427** |
| CrossAD | 0.3115 | **0.9320** | 0.4135 | **0.9386** | 0.5904 | **0.8628** | 0.4871 | **0.7857** | 0.1829 | **0.5374** | 0.2973 | **0.9511** | 0.3383 | **0.7412** |
| DLinear | 0.4549 | **0.9384** | 0.5654 | **0.9432** | 0.7728 | **0.8615** | 0.6728 | **0.8437** | 0.2268 | **0.4402** | 0.4283 | **0.9542** | 0.4782 | **0.7325** |
| KANAD | 0.3785 | **0.9465** | 0.5360 | **0.9157** | 0.6223 | **0.8628** | 0.6666 | **0.8491** | 0.2066 | **0.3936** | 0.2905 | **0.9543** | 0.4409 | **0.7279** |
| LSTMAE | 0.3265 | **0.9463** | 0.5501 | **0.9460** | 0.6798 | **0.8629** | 0.6183 | **0.7770** | 0.2193 | **0.4329** | 0.4677 | **0.9521** | 0.4074 | **0.7328** |
| MICN | 0.9141 | **0.9406** | 0.4898 | **0.9150** | 0.7545 | **0.8708** | 0.6915 | **0.7856** | 0.2085 | **0.4307** | 0.6111 | **0.9575** | 0.6402 | **0.7540** |
| ModernTCN | 0.5646 | **0.9331** | 0.3545 | **0.9177** | 0.6903 | **0.8724** | 0.7321 | **0.8262** | 0.1850 | **0.4614** | 0.2495 | **0.9643** | 0.5369 | **0.7452** |
| TimeMixer++ | 0.7936 | **0.9356** | 0.5179 | **0.9450** | 0.7018 | **0.8661** | 0.3764 | **0.4063** | 0.1889 | **0.5374** | 0.2946 | **0.9509** | 0.6232 | **0.7390** |
| TimesNet | 0.8417 | **0.9342** | 0.4868 | **0.9171** | 0.7828 | **0.8672** | 0.5289 | **0.8301** | 0.1961 | **0.5249** | 0.5873 | **0.9601** | 0.6505 | **0.7619** |
| **AVG** | 0.5475 | **0.9388** | 0.4794 | **0.9308** | 0.6804 | **0.8668** | 0.6001 | **0.7729** | 0.1997 | **0.4577** | 0.4026 | **0.9557** | 0.5095 | **0.7419** |

based metrics (Aff-F1). By effectively filtering out non-structural background noise, COGNOS amplifies the signal-to-noise ratio for genuine anomalous events. These results indicate that COGNOS successfully disentangles true structural deviations from the high-frequency fluctuations that typically plague MSE-based scoring, thereby minimizing false positives in complex industrial signals.

Range-based metrics (e.g., VUS, R-AUC) also exhibit consistent improvements, the magnitude of these gains is more moderate compared to event-based scores. There are instances where metrics decrease (e.g., TimesNet on MSL and KANAD on PSM). This phenomenon is theoretically consistent with the dynamics of our ARKS module. For long-duration anomalies, once the filter's state adapts to the new structural level (where $y_{t+1} \approx y_t$), the innovation term rapidly vanishes ($\tilde{y}_{t+1} \approx 0$), causing the anomaly score to stabilize.[4] Consequently, COGNOS is designed to be hypersensitive to the **anomaly onset** rather than passively tracking prolonged steady-state shifts. This characteristic is highly desirable in real-world deployments, where early detection of the structural break is paramount for timely intervention. This score decay directly penalizes range-based overlap metrics that reward full temporal coverage. Applications that require precise anomaly duration tracking should consider this trade-off; see Section 6 for a formal discussion.

**Ablation Studies.** To verify the specific contributions of each component and their theoretical coupling, we conducted a component-wise analysis using the KAN-AD backbone. The results, reported in terms of Std-F1 and Aff-F1 in Table 3, validate the necessity of the proposed synergy.

**Adaptive Residual Kalman Smoother:** We first validate our ARKS module by replacing it with two common filtering alternatives: a heuristic moving average (MA) and a classic low-pass filter (LP). The performance degradation

---

[4]Analysis detailed in Appendix A.3.3.

*Table 3.* Ablation Study of COGNOS Components and Filtering Methods. The best results are highlighted in **bold**, and the second-best are underlined. The full experiment results and more details can be found in the Appendix D.3.

| Datasets | | | | GECCO | | MSL | | PSM | |
|---|---|---|---|---|---|---|---|---|---|
| GWNR | ARKS | MA | LP | Std-F1 | Aff-F1 | Std-F1 | Aff-F1 | Std-F1 | Aff-F1 |
| ✗ | | | | 0.4738 | 0.3785 | 0.8115 | 0.5360 | 0.8946 | 0.6223 |
| ✗ | ✓ | | | 0.4740 | 0.4845 | 0.8083 | 0.5285 | 0.9433 | 0.6689 |
| ✓ | | | | 0.3440 | 0.4098 | 0.8704 | 0.5917 | 0.9160 | 0.4910 |
| ✓ | | ✓ | | 0.3531 | 0.4123 | 0.1626 | 0.2564 | 0.7979 | 0.1845 |
| ✓ | | | ✓ | 0.3539 | 0.4236 | 0.1662 | 0.2584 | 0.8006 | 0.2302 |
| ✓ | ✓ | | | **0.7466** | **0.9465** | **0.9165** | **0.9157** | **0.9744** | **0.8628** |

observed with both alternatives underscores the superiority of our statistically-grounded approach. While heuristic filters can reduce noise, they suppress noise at the cost of blurring the onset of anomalies, while the Kalman Smoother is theoretically optimal because our GWNR creates the ideal Gaussian white noise conditions under which it can effectively separate the true anomaly signal from random fluctuations. Comparing ARKS alone against the full COGNOS framework further reveals a critical dependency: without whitened residuals, ARKS achieves only modest and inconsistent gains (PSM Std-F1: 0.8946→0.9433, but MSL Std-F1 marginally declines from 0.8115 to 0.8083), whereas the full COGNOS achieves reliable improvements across all datasets. This confirms that statistical rigor in the residuals is a prerequisite for the Kalman smoother to approach its theoretical optimality.

**Gaussian-White Noise Regularization:** Removing GWNR degrades performance consistency, underscoring its foundational role in compelling statistically random residuals to prevent overfitting to spurious patterns and ensure a robust normalcy representation. A particularly revealing case is GECCO: applying GWNR alone *without* ARKS *decreases* Std-F1 from 0.4738 to 0.3440. By redistributing residual energy toward a flatter, Gaussian-like distribution, GWNR inadvertently reduces the direct discriminability of

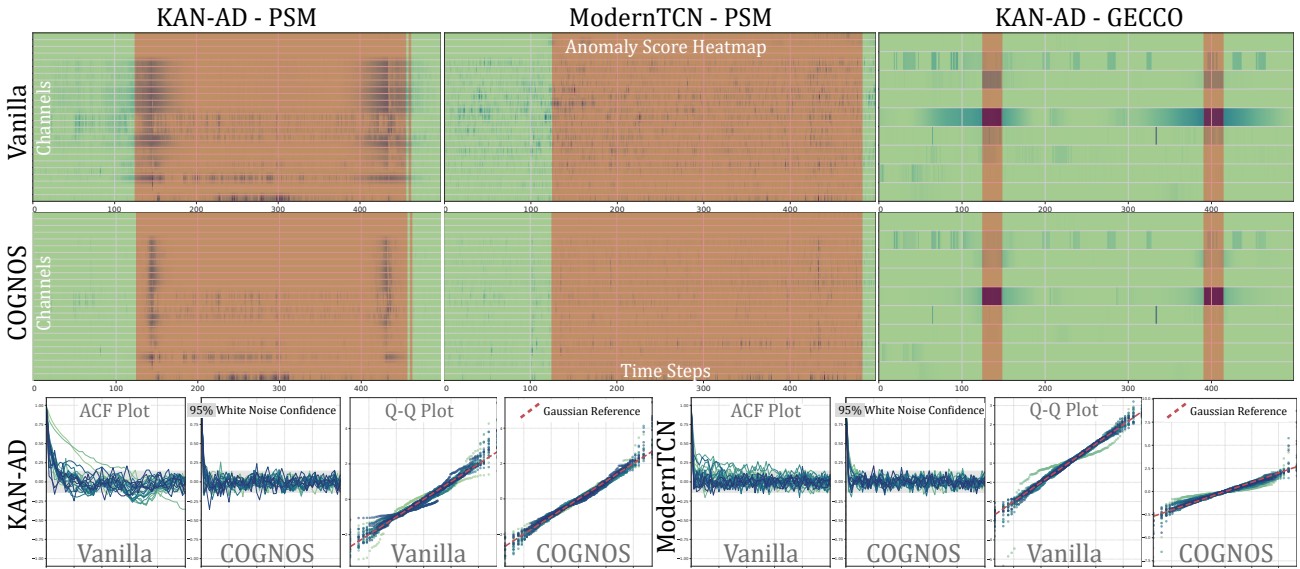

*Figure 3.* Qualitative Analysis on Signal Purity and Statistical Physics. **Top Row:** Anomaly score dynamics on GECCO and PSM datasets. **Bottom Row:** Statistical diagnostics of reconstruction residuals.

the raw residual magnitude — precisely the condition that necessitates ARKS to recover selectivity through hypothesis-driven state estimation. The complete COGNOS's superior performance confirms the powerful synergy: GWNR creates the statistical preconditions, and ARKS provides the selectivity to translate them into reliable detection.

*Table 4.* Computational Efficiency Analysis. We report the **average execution time (ms) per iteration** (Training) and **per sample point** (Inference), (+) denotes the additional overhead introduced by our COGNOS framework. We set batch size $= 128$ and sequence length $= 128$

| **Datasets** | **Models** | Autoformer | | KANAD | | TimesNet | |
|---|---|---|---|---|---|---|---|
| | | Vanilla | COGNOS | Vanilla | COGNOS | Vanilla | COGNOS |
| **GECCO** | Train | 62.53 | +41.648 | 39.94 | +48.4 | 101.34 | +171.49 |
| | Inference | 0.1782 | +0.1274 | 0.0847 | +0.1467 | 0.1694 | +0.1513 |
| **PSM** | Train | 53.30 | +82.66 | 97.31 | +206.68 | 103.89 | +257.55 |
| | Inference | 0.2150 | +0.1031 | 0.2459 | +0.1387 | 0.2054 | +0.1327 |
| **SWAN** | Train | 87.64 | +164.11 | 216.54 | +355.01 | 153.57 | +359.27 |
| | Inference | 0.2482 | +0.0838 | 0.3546 | +0.1268 | 0.2190 | +0.1402 |

**Efficiency.** As detailed in Table 4, we analyze the computational impact of our framework. **Inference:** Attributed to the recursive nature of the ARKS filter, the framework acts as a stream-processing compatible module, introducing a negligible average latency of approximately **0.13 ms/point**. **Training:** The GWNR loss incurs an average training overhead of 187 ms/iteration, primarily driven by the spectral and kernel-based computations. However, this cost is strictly confined to the **offline phase**. It is also worth noting that the total training duration with COGNOS often remains comparable to the intrinsic variability observed between different backbone architectures (e.g., the gap between KAN-AD and TimesNet), rendering it a practical solution given the significant performance gains.

**Visualizations.** Fig. 3 confirms COGNOS's effectiveness. **Top Row**: baseline anomaly scores are contaminated by non-anomalous high-frequency noise masking subtle anomalies; COGNOS produces a cleaner signal with suppressed background noise and prominent peaks at true anomaly onset. **Bottom Row**: baseline residuals violate statistical assumptions (temporal autocorrelation/non-Gaussianity), while GWNR-moderated residuals are closer to Gaussian white noise (suppressed autocorrelation, tighter Q-Q alignment). This confirms our regularization strategy engineers desired residual properties, the key reason for cleaner, reliable scores. Appendix C includes more visualizations.

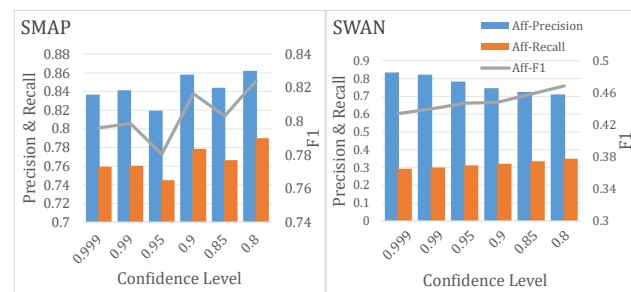

*Figure 4.* The impact of the Confidence Level $(1-\alpha)$ on KAN-AD backbone across the SMAP and SWAN datasets.

**Hyperparameter Sensitivity.** A key advantage of COGNOS is its minimal configuration requirement, relying on a single, statistically interpretable hyperparameter: the Confidence Level $(1-\alpha)$ for the ARKS Circuit Breaker. We evaluate the sensitivity of this parameter on KAN-AD backbones (Fig. 4) across a wide range of $(1-\alpha) \in \{0.999, \dots, 0.8\}$. The results demonstrate that COGNOS is robust to hyperparameter variations, maintaining high performance across a broad range. Lower confidence levels $(< 0.9)$ make the Circuit Breaker more aggressive, increasing sensitivity to subtle

anomalies but potentially admitting more noise. Conversely, extremely high levels ($> 0.9$) favor strong smoothing, prioritizing precision. We also observed that this sensitivity is notably attenuated in multivariate datasets (e.g., SWAN) compared to univariate ones (e.g., SMAP), indicating that the offline calibration possesses superior robustness in multivariate scenarios.

## 6. Conclusion

In this work, we propose COGNOS, which addresses a core limitation of reconstruction-based time series anomaly detection: the statistically flawed and noisy anomaly scores generated by standard MSE-based training and inference. Experiments demonstrate that COGNOS is model-agnostic and consistently effective across diverse architectures and datasets. Our results confirm this improvement arises from the strong synergy between our regularization and smoothing components. Furthermore, our work shows that directly regularizing output statistics is a powerful strategy, presenting direction beyond conventional architectural or representation-focused improvements.

Beyond empirical gains, our work establishes **residual engineering** as a principled axis of improvement that is orthogonal and complementary to architectural innovation. While the field has largely focused on increasingly complex encoder designs, we show that enforcing the statistical quality of reconstruction residuals is a largely untapped yet highly effective strategy — achievable without modifying backbone architecture or inference protocol.

Several directions remain open. Extending ARKS to handle state-dependent observation noise (i.e., a time-varying $R$ matrix) would broaden applicability to heteroscedastic sensors. Developing analogous output-residual constraints for non-MSE objectives, such as contrastive or association-discrepancy losses would extend the framework beyond its current applicability boundary and represents a natural continuation of this line of research.

## Limitations

While COGNOS demonstrates broad effectiveness, its design rests on assumptions that define a well-scoped operating regime; we discuss three boundaries below and outline directions for extending the framework beyond them.

**Temporal Heteroscedasticity.** The ARKS module calibrates a fixed observation noise covariance matrix $R$ from the training set, implicitly assuming that the sensor noise floor remains stable across operational states. This assumption holds well for cyber-physical and industrial systems such as SWaT and MSL, where noise is driven primarily by thermal and electronic factors rather than signal magni-

tude. In applications where noise variance scales with the system state — such as proportional error models common in biomedical sensing — a fixed $R$ becomes sub-optimal and may trigger false alarms during high-signal periods. Extending ARKS to a state-dependent, adaptive $R$ estimation scheme is a natural direction for future work.

**Applicability Boundary.** COGNOS is designed specifically for backbone models trained with MSE-based reconstruction objectives, as its core assumptions — that normal residuals approximate Gaussian white noise and that anomalies manifest as structural breaks from this distribution — are grounded in the MSE-as-MLE interpretation. Models that optimize alternative criteria, such as the Anomaly Transformer (Xu et al., 2022), which maximizes Association Discrepancy rather than minimizing reconstruction error, do not produce the residual signals that ARKS is designed to post-process, and therefore fall outside the applicable scope of our framework. Extending the residual engineering principle to such alternative objectives represents a promising avenue for broadening the framework's reach; further comparisons with standalone TSAD baselines are provided in Appendix D.5.

**Onset Detection vs. Coverage.** The Circuit Breaker mechanism is deliberately tuned to maximize sensitivity to anomaly onsets while suppressing sustained alarm fatigue arising from long steady-state deviations. This design is well-suited to safety-critical early-warning scenarios — such as fault detection in water treatment (SWaT) or spacecraft telemetry (MSL) — where rapid identification of the structural break is paramount for timely intervention. Applications requiring precise anomaly duration tracking or full-segment coverage annotation, where range-based metrics that reward temporal overlap (e.g., R-A-R, V-R) are the primary criterion, may find this onset bias a disadvantage; in such settings, a more conservative Circuit Breaker threshold may offer a better operating point.

## Impact Statement

This paper presents work whose goal is to advance the field of Machine Learning. There are many potential societal consequences of our work, none which we feel must be specifically highlighted here.

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

# A. Theoretical Analysis

## A.1. Analysis of Mean Squared Error Reconstruction Objectives

In this section, we provide a formal derivation connecting the standard Mean Squared Error (MSE) objective to the Maximum Likelihood Estimation (MLE) under specific probabilistic assumptions. We then discuss how violations of these assumptions in industrial scenarios motivate our proposed approach.

### A.1.1. PROBABILISTIC INTERPRETATION OF MSE

**Proposition A.1.** *Minimizing the Mean Squared Error (MSE) is mathematically equivalent to maximizing the likelihood of the data, assuming the reconstruction residuals follow an independent and identically distributed (i.i.d.) Gaussian distribution with zero mean and fixed variance.*

*Proof.* Let $\mathcal{D} = \{(\mathbf{x}_i, y_i)\}_{i=1}^{N}$ denote a dataset of $N$ i.i.d. samples. Consider a parametric model $f_\theta : \mathcal{X} \to \mathcal{Y}$ parameterized by $\theta$. We assume the observed target $y$ is generated by the deterministic function $f_\theta(\mathbf{x})$ perturbed by additive noise $\epsilon$ (The Wold Decomposition Hypothesis):

$$y_i = f_\theta(\mathbf{x}_i) + \epsilon_i, \tag{17}$$

where the noise term follows a **Gaussian distribution** $\epsilon_i \sim \mathcal{N}(0, \sigma^2)$ with fixed variance $\sigma^2$.

The probability density function (PDF) for a single observation $y_i$ given input $\mathbf{x}_i$ is:

$$p(y_i|\mathbf{x}_i; \theta) = \frac{1}{\sqrt{2\pi\sigma^2}} \exp\left(-\frac{(y_i - f_\theta(\mathbf{x}_i))^2}{2\sigma^2}\right). \tag{18}$$

Assuming the samples are independent, the likelihood function for the entire dataset is the product of individual densities:

$$\mathcal{L}(\theta) = \prod_{i=1}^{N} p(y_i|\mathbf{x}_i; \theta) = \prod_{i=1}^{N} \frac{1}{\sqrt{2\pi\sigma^2}} \exp\left(-\frac{(y_i - f_\theta(\mathbf{x}_i))^2}{2\sigma^2}\right). \tag{19}$$

To simplify the optimization, we consider the log-likelihood $\ell(\theta) = \log \mathcal{L}(\theta)$:

$$\begin{aligned}
\ell(\theta) &= \sum_{i=1}^{N} \log\left(\frac{1}{\sqrt{2\pi\sigma^2}}\right) - \sum_{i=1}^{N} \frac{(y_i - f_\theta(\mathbf{x}_i))^2}{2\sigma^2} \\
&= -\frac{N}{2}\log(2\pi\sigma^2) - \frac{1}{2\sigma^2}\sum_{i=1}^{N}(y_i - f_\theta(\mathbf{x}_i))^2.
\end{aligned} \tag{20}$$

Since $N$ and $\sigma^2$ are constants with respect to $\theta$, maximizing the log-likelihood is equivalent to minimizing the negative term:

$$\theta_{\text{MLE}}^* = \underset{\theta}{\operatorname{argmax}}\, \ell(\theta) \iff \underset{\theta}{\operatorname{argmin}} \sum_{i=1}^{N}(y_i - f_\theta(\mathbf{x}_i))^2. \tag{21}$$

The term $\sum(y_i - f_\theta(\mathbf{x}_i))^2$ corresponds to the Sum of Squared Errors (SSE), which is proportional to the Mean Squared Error (MSE). Thus, minimizing MSE yields the Maximum Likelihood Estimate under the assumption of homoscedastic Gaussian noise. □

### A.1.2. LIMITATIONS IN INDUSTRIAL ANOMALY DETECTION

The derivation in Proposition A.1 highlights two strong inductive biases inherent in MSE-based reconstruction methods, which are often violated in real-world industrial data:

1. **Assumption of Gaussianity (Light Tails):** The term $(y_i - f_\theta(\mathbf{x}_i))^2$ imposes a quadratic penalty, implicitly assuming that the probability of large errors decays exponentially (thin tails). Industrial data often exhibits heavy-tailed distributions due to transient spikes or sensor glitches. MSE is notoriously sensitive to outliers, forcing the model to "memorize" anomalies to reduce the high quadratic penalty, thereby degrading the reconstruction quality of normal patterns.

2. **Assumption of Homoscedasticity (Constant Variance):** The derivation assumes $\sigma^2$ is constant across all samples (homoscedasticity). Industrial data are inherently heteroscedastic. For instance, a machine operating at high load may exhibit naturally higher vibration variance compared to an idle state. An MSE-based model treats all deviations equally, causing it to over-penalize high-variance regions while potentially under-penalizing subtle anomalies in low-variance regions.

These limitations necessitate the use of GWNR Loss to capture the complex stochastic nature of the data.

## A.2. Optimality of Kalman Filtering under Gaussian Assumptions

We provide the theoretical justification for the Gaussianity Constraint imposed by the GWNR loss (specifically $\mathcal{L}_{\mathrm{MMD}}$) and demonstrate that enforcing the residual distribution to be Gaussian is necessary to elevate the downstream Adaptive Residual Kalman Smoother (ARKS) from a merely "best linear" estimator to the "globally optimal" estimator.

### A.2.1. PROBLEM SETUP

Consider the Linear Gaussian State Space Model (LG-SSM) used in ARKS. For a simplified scalar case [5], the system is defined as:

$$
\begin{cases}
x_t = A x_{t-1} + w_t, & w_t \sim p_w(w) \\
y_t = H x_t + v_t, & v_t \sim p_v(v)
\end{cases},
\tag{22}
$$

where $x_t$ is the latent signal (anomaly/trend), $y_t$ is the observed residual from the backbone model and $A$ governs the temporal evolution of the trend (e.g., $A = 1$ for a random walk). , we set the observation matrix $H = 1$. This implies that the latent state is directly observable subject to additive noise, without further projection or scaling.

The objective of the filter is to compute the estimate $\hat{x}_t$ given observations $Y_t = \{y_1, \ldots, y_t\}$ that minimizes the Mean Squared Error (MSE):

$$
J(\hat{x}_t) = \mathbb{E}\left[(x_t - \hat{x}_t)^2 \mid Y_t\right].
\tag{23}
$$

### A.2.2. OPTIMALITY ANALYSIS

**Proposition A.2.** *The estimator that minimizes the MSE for any arbitrary distribution is the Conditional Expectation:*

$$
\hat{x}_{MMSE} = \mathbb{E}[x_t \mid Y_t].
\tag{24}
$$

**Case 1: Non-Gaussian Noise (Standard Reconstruction).** If the noise terms $w_t$ and $v_t$ follow arbitrary non-Gaussian distributions, the conditional density $p(x_t|Y_t)$ becomes complex and non-Gaussian. In this scenario:

- The optimal MMSE estimator $\mathbb{E}[x_t|Y_t]$ is generally a **non-linear** function of the observations $Y_t$.

- The Kalman Filter, which is strictly a linear recursive operator, corresponds to the Best Linear Unbiased Estimator.

- Consequently, there exists a strictly positive performance gap: $J(\hat{x}_{\mathrm{KF}}) > J(\hat{x}_{\mathrm{MMSE}})$.

**Case 2: Gaussian Noise (COGNOS Condition).** To bridge the gap between linear estimation and global optimality, we must address the distributional properties of both noise sources: the measurement noise $v_t \sim p_v$ and the process noise $w_t \sim p_w$. The GWNR loss minimizes the Maximum Mean Discrepancy ($\mathcal{L}_{\mathrm{MMD}}$) between the reconstruction residuals and a standard Gaussian distribution. Since the training data consists of normal sequences where structural anomalies are absent

---

[5]*Remark.* The derivation above utilizes a scalar model formulation. For multivariate time series with $F$ channels, the global MMSE estimator corresponds to a multivariate Kalman Filter operating on the full covariance matrix $\Sigma \in \mathbb{R}^{F \times F}$. However, multivariate KF incurs a cubic computational cost $O(F^3)$, which is prohibitive for high-dimensional industrial data. Our proposed ARKS employs a channel-wise factorization, which is mathematically equivalent to the multivariate KF if and only if the noise covariance matrices are diagonal (i.e., residuals are spatially independent). This relies on the inductive bias that the multivariate backbone $f_\theta$ effectively minimizes mutual information between residual channels by capturing all cross-series correlations in the deterministic trend. Under this spatial Whitening assumption, the channel-wise optimality derived above extends to the multivariate system without loss of generality.

(i.e., latent state $x_t = 0$), the observed residual $y_t$ is equivalent to the measurement noise $v_t$. Thus, by minimizing $\mathcal{L}_{\text{MMD}}$, COGNOS explicitly **enforces** $p_v(v)$ to approximate a Gaussian distribution:

$$\min \mathcal{L}_{\text{MMD}}(y_t, \mathcal{N}) \implies v_t \xrightarrow{d} \mathcal{N}(0, R). \tag{25}$$

The process noise $w_t$ regulates the temporal evolution of the latent trend $x_t$, which represents the deterministic dynamics leaked from the backbone inference process (as analyzed in Sec. 4.2, Mode I). Unlike $v_t$, the distribution of this leakage evolution cannot be regularized by GWNR directly. Instead, the Gaussianity of $w_t$ is a **modeling assumption** inherent to our choice of Random Walk or AR(1) priors for tracking smooth spectral leakage, this choice corresponds to the **maximum entropy** principle for a fixed variance. Accordingly, we assume $w_t$ follows a Gaussian distribution as our prior.

## A.3. Analysis of the Circuit Breaker

In this section, we provide a rigorous derivation of the Circuit Breaker's dynamic behavior. We model the activation of the mechanism as an impulsive injection of process noise and analyze the system's response across two distinct phases: the **Trigger Phase** ($t$) and the **Recovery Phase** ($t + 1$).

### A.3.1. PHASE I: THE TRIGGER EVENT (AT TIME $t$)

Consider the anomaly detection at time $t$ where the NIS score $\epsilon_t$ exceeds the threshold $\tau_\alpha$. The mechanism inflates the process noise covariance $Q_t^* = \beta R$ with $\beta \gg 1$. We analyze the asymptotic behavior of the Kalman Gain $K_t$ and the posterior covariance $P_{t|t}$ as $\beta \to +\infty$.

**Proposition A.3** (Memory Erasure Property). *As the inflation factor $\beta \to +\infty$, the filter instantaneously forgets prior information, converging to a memoryless estimator.*

The predicted covariance $P_{t|t-1}^* = A^2 P_{t-1|t-1} + \beta R$ becomes dominated by the inflation term. The Kalman Gain is given by:

$$\lim_{\beta \to +\infty} K_t = \lim_{\beta \to +\infty} \frac{P_{t|t-1}^*}{P_{t|t-1}^* + R} = \lim_{\beta \to +\infty} \frac{P_{prior} + \beta R}{P_{prior} + (\beta + 1)R} = 1. \tag{26}$$

Consequently, the state estimate updates as $\hat{x}_{t|t} = \hat{x}_{t|t-1} + 1 \cdot (y_t - \hat{x}_{t|t-1}) = y_t$. This proves that the filter achieves zero-lag tracking during the anomaly, treating the current observation as the true state.

Crucially, the mechanism resets the system's uncertainty. Using the Joseph form for numerical stability, the posterior covariance converges to:

$$\lim_{\beta \to +\infty} P_{t|t} = \underbrace{(1 - K_t)^2 P_{t|t-1}^*}_{\to 0} + \underbrace{K_t^2 R}_{\to R} = R. \tag{27}$$

This result ($P_{t|t} = R$) signifies that strictly after a trigger event, the uncertainty of the state estimate is exactly equal to the measurement noise variance.

### A.3.2. PHASE II: THE RECOVERY DYNAMICS (AT TIME $t + 1$)

At step $t + 1$, we assume the external anomaly trigger is removed, and the process noise returns to its nominal value $Q_{t+1} = Q_{\text{base}}$. However, the system state evolves from the reset covariance derived in Eq. 27.

The predicted covariance for $t + 1$ is:

$$P_{t+1|t} = A^2 P_{t|t} + Q_{\text{base}} \approx A^2 R. \tag{28}$$

Here, we assume $Q_{\text{base}} \ll R$ (smooth prior) is negligible compared to the propagated uncertainty $A^2 R$.

The Kalman Gain for the recovery step stabilizes at a deterministic value dependent solely on the channel dynamics $A$:

$$K_{t+1} = \frac{P_{t+1|t}}{P_{t+1|t} + R} \approx \frac{A^2 R}{A^2 R + R} = \frac{A^2}{1 + A^2}. \tag{29}$$

*Remark.* For a highly persistent channel (e.g., a Random Walk where $A \approx 1$), $K_{t+1} \approx 0.5$. This contrasts sharply with the steady-state smoothing mode where $K_{ss} \approx 0$. It implies that immediately after an anomaly, the filter enters a "Transient Verification State," assigning equal weight to the model prediction and the new observation.

This paper introduces a universal method that improves reconstruction-based time series anomaly detection by forcing reconstruction errors to behave like ideal Gaussian White noise, and uses an adaptive Kalman smoother to extract stable anomaly during inference.

### A.3.3. SENSITIVITY ANALYSIS VIA NIS SCALING

We now investigate whether the system is prone to false alarms at $t + 1$. The detection statistic is the NIS score: $\epsilon_{t+1} = \tilde{y}_{t+1}^2/S_{t+1}$, where $\tilde{y}_{t+1} = (y_{t+1} - \hat{x}_{t+1|t})$.

The innovation covariance $S_{t+1}$ is expanded due to the uncertainty reset:

$$S_{t+1} = P_{t+1|t} + R \approx (1 + A^2)R. \tag{30}$$

Compared to the steady-state variance $S_{ss} \approx R$, the detection threshold is effectively relaxed by a factor of $(1 + A^2)$.

We analyze the NIS score under two anomaly scenarios:

**Case 1: Step Anomaly.** If the anomaly persists ($y_{t+1} \approx y_t$), the prediction $\hat{x}_{t+1|t} = A\hat{x}_{t|t} \approx Ay_t$ aligns with the observation. The innovation $\tilde{y}_{t+1} \approx y_{t+1} - Ay_t \approx (1 - A)y_t$. For $A \approx 1$, $\tilde{y}_{t+1} \to 0$, resulting in $\epsilon_{t+1} \approx 0$. The mechanism correctly accepts the new level as normal.

**Case 2: Point Anomaly.** If the signal returns to zero ($y_{t+1} \approx 0$) after a spike at $y_t$, a "rebound" innovation occurs: $\tilde{y}_{t+1} \approx 0 - Ay_t$. A standard filter might flag this large negative residual as a secondary anomaly. However, ARKS mitigates this via the variance inflation. The NIS score becomes:

$$\epsilon_{t+1} = \frac{(-Ay_t)^2}{(1 + A^2)R} = \underbrace{\left(\frac{A^2}{1 + A^2}\right)}_{\eta < 1} \frac{y_t^2}{R} \approx \eta \cdot \epsilon_t^{raw}, \tag{31}$$

whrere $\epsilon_t^{\text{raw}} \triangleq y_t^2/R$ as the standardized magnitude of the initial anomaly at time $t$ (under the steady-state approximation $S_t \approx R$). For $A = 1$, $\eta = 0.5$. The score is effectively halved. Even if the rebound residual is large, the Circuit Breaker implies that for a secondary alarm to trigger at $t + 1$, the "rebound" magnitude must be significantly larger (approx. $\sqrt{2}$ times) than the initial anomaly logic suggests. This creates a theoretical "Refractory Period", drastically reducing False Positives caused by transient recovery.

Algorithm 1 outlines the procedure. To stabilize the input for ARKS, we employ a Fixed-Buffer Aggregator. Specifically, for any time step $\tau$, the residual $y_\tau$ is computed by averaging the predictions from all $w$ overlapping windows that cover $\tau$, finalized after a buffer of $k$ steps. This ensemble strategy ensures that the input $y_\tau$ to the Kalman filter minimizes reconstruction variance, allowing the Circuit Breaker to focus solely on structural anomalies. In all experiments, we set $k = w - 1$, where $w$ is the window length of the backbone, so that each timestamp receives contributions from all overlapping windows before ARKS updates. The parameter $k \in (0, w-1]$ is adjustable: smaller values reduce detection latency at the cost of numerical stability, while $k = w-1$ maximizes stability and is recommended for latency-insensitive deployments.

---

**Algorithm 1** Streaming Anomaly Detection with ARKS

---

**Input** Data Stream $\{\mathbf{x}_t\}$, Model $\mathcal{M}$, Buffer $k$, Threshold $\tau_\alpha$
**Output** Anomaly Scores $\{\mathcal{S}_t\}$
 1: **Initialization Phase:**
 2: *// Determine parameters using a calibration set*
 3: Initialize system matrices $\{A, Q_{\text{base}}, R\}$ (Mode I/II)
 4: Initialize state $\hat{x}_{0|0}$, covariance $P_{0|0}$ and Aggregator $\mathcal{A}$.
 5: **for** $t = 1, 2, \ldots$ **do**
 6:    $\hat{\mathbf{W}}_t \leftarrow \mathcal{M}(\mathbf{x}_{t-w+1:t})$ {Reconstruct current window}
 7:    Update $\mathcal{A}$ with $\hat{\mathbf{W}}_t$ {Accumulate overlapping predictions}
 8:    *// Inference at time $\tau$*
 9:    $\tau \leftarrow t - k$
10:    **if** $\tau \geq 1$ **then**
11:       Retrieve aggregated residual $y_\tau \leftarrow \mathbf{x}_\tau - \text{Mean}(\mathcal{A}, \tau)$
12:       **[Step 1] Time Update:**
13:       $\hat{x}_{\tau|\tau-1} = A\hat{x}_{\tau-1|\tau-1}$
14:       **[Step 2] Circuit Breaker:**
15:       Compute NIS score $\epsilon_\tau = (y_\tau - \hat{x}_{\tau|\tau-1})^2 / (P_{\tau|\tau-1} + R)$
16:       **if** $\epsilon_\tau > \tau_\alpha$ **then**
17:          $Q_\tau^* \leftarrow \beta R$
18:       **else**
19:          $Q_\tau^* \leftarrow Q_{\text{base}}$
20:       **end if**
21:       Update prior covariance $P_{\tau|\tau-1}^*$ using $Q_\tau^*$
22:       **[Step 3] Measurement Update & Scoring:**
23:       Compute $K_\tau$, update posterior $\hat{x}_{\tau|\tau}$ using Eq. 16
24:       $\mathcal{S}_\tau \leftarrow \|\hat{x}_{\tau|\tau}\|^2$
25:    **end if**
26: **end for**
27: **return** $\mathcal{S}_{1:T}$

---

## B. Dataset Details

The the datasets used in the experiments are as follows, details are in Table 5:

*Table 5.* Dataset Details

| Dataset | Category | Dimension | Training | Validation | Test | AR (%) |
|---------|----------|-----------|----------|------------|------|--------|
| GECCO | Water treatment | 9 | 55,408 | 13,852 | 69,261 | 1.25 |
| MSL | Spacecraft | 55 | 46,653 | 11,664 | 73,729 | 10.5 |
| SMAP | Spacecraft | 1 | 108,146 | 27,037 | 427,617 | 12.8 |
| PSM | Server Machine | 25 | 105,984 | 26,497 | 87,841 | 27.8 |
| SWAN | Space Weather | 38 | 48,000 | 12,000 | 60,000 | 23.8 |
| SWaT | Water treatment | 51 | 396,000 | 99,000 | 449,919 | 12.1 |
| UCR | Natural | 1 | 1,790,680 | 447,670 | 6,143,541 | 0.6 |

- **GECCO** (Moritz et al., 2018) is a real-world water quality monitoring dataset provided by Thüringer Fernwasserversorgung (Thuringian Central Water Supply) in collaboration with the IMProvT research project. Originating from the anomaly detection competition at the Genetic and Evolutionary Computation Conference (GECCO), this benchmark challenges algorithms to identify subtle, undesirable variations in environmental sensor streams while maintaining exceptionally low false alarm rates for critical requirement of operational water safety systems. The dataset serves as a rigorous testbed for evaluating anomaly detection methodologies in environmentally sensitive infrastructure.

- **MSL & SMAP** (Hundman et al., 2018) utilize expert-labeled Incident Surprise Anomaly (ISA) reports from two operational domains: the Mars Science Laboratory (MSL) rover Curiosity and the Soil Moisture Active Passive (SMAP) satellite. These reports constitute canonical post-launch documentation of unexpected events that may jeopardize spacecraft health, serving as authoritative ground-truth data for anomaly-driven risk assessment. The objective is to leverage these curated corpora to advance data-driven methods for operational risk identification in robotic space missions.

- **PSM (Pooled Server Metrics)** (Abdulaal et al., 2021) is a publicly released, anonymized dataset derived from internal monitoring traces of multiple application-server nodes at eBay. Its primary research purpose is to supply a large-scale, real-world resource-usage corpus for the development and evaluation of anomaly-detection methodologies in distributed production environments.

- **SWAN (Space Weather Analytics)** (Angryk et al., 2020) is a benchmark dataset for solar flare prediction, constructed from Space Weather HMI Active Region Patches provided by the Joint Science Operations Center. It integrates cleaned, multi-source verified time-series of solar vector magnetograms captured by NASA's Solar Dynamics Observatory Helioseismic Magnetic Imager between May 2010 and August 2018. This standardized resource enables unbiased evaluation of machine learning models for detecting solar eruptions that threaten astronauts, spacecraft systems, and terrestrial power infrastructure.

- **SWaT (Secure Water Treatment)** (Mathur & Tippenhauer, 2016) is a purpose-built, small-scale yet fully operational water-treatment testbed designed to mirror the physical processes and control architectures of contemporary municipal plants. Jointly specified with Singapore's Public Utility Board and realized by an industrial vendor, the facility serves as a realistic cyber-physical research platform. Its primary scientific objective is to systematically investigate cyber-attack mechanisms and plant-level responses, and to experimentally validate novel, especially physics-based, countermeasures.

- **UCR (UCR Time Series Anomaly Archive)** (Wu & Keogh, 2023) is a rigorously curated benchmark collection designed to overcome critical flaws in existing time series anomaly detection datasets. Created to enable reliable algorithm comparisons and meaningful progress assessment, this archive spans diverse domains including medicine, sports, entomology, industry, space science, and robotics. The datasets emphasize high-quality ground truth with predominantly single anomalies per series, while intentionally incorporating a spectrum of difficulties ranging from trivial cases detectable via simple heuristics (e.g., abrupt sensor dropouts from missing data artifacts) to highly complex patterns. Serving as a community standard analogous to the UCR Time Series Classification Archive, it provides metadata-rich, vetted resources to foster reproducible research and robust methodological advances in time series anomaly detection.

**Implementation Details.** To ensure a fair and reproducible comparison, we strictly prioritized the default hyperparameter configurations provided in the official open-source implementations for all backbone models, with detailed settings listed in Table 6. Regarding our proposed framework, COGNOS introduces a single user-defined hyperparameter, the Circuit Breaker Confidence Level $(1 - \alpha)$, which was universally set to $0.90$ across all datasets to demonstrate the method's robustness without intensive tuning, we used the training set as the calibration set for ARKS. Minor adjustments were made solely for certain memory-intensive architectures (e.g., CrossAD, TimesNet, TimeMixer++) to accommodate hardware constraints (24GB VRAM), while ensuring that training stability and convergence behaviors remained consistent with the original literature.

*Table 6.* Hyperparameters Configurations

**Miscellaneous Configurations**

| Datasets | Hyper-parameter | | | | | | Traning Process | | |
|---|---|---|---|---|---|---|---|---|---|
| | Window Length | dModel | dMLP | Layers | Num Heads | Confidence | Learning Rate[6] | Batch Size | Epochs |
| GECCO | | | | | | | | | 10 |
| UCR | 128 | | | | | | | | 3 |
| SWaT | | | | | | | | | 10 |
| MSL | 96 | 128 | 128 | 3 | 8 | 0.9 | 1e-4 | 128 | 3 |
| SWAN | | | | | | | | | 10 |
| PSM | 192 | | | | | | | | 3 |
| SMAP | | | | | | | | | |

**Model Specific Hyper-parameter**

**CrossAD**

| Datasets | Window Length | dMLP | Num Heads | Patch Length | MS kernels | MS Method | Bank Size | Query Length | Topk | M Layers | E Layers | D Layers |
|---|---|---|---|---|---|---|---|---|---|---|---|---|
| GECCO | 128 | | | | | | | | | | | |
| UCR | | | | | | | | | | | | |
| MSL | | | | | | | | | | | | |
| SWaT | 96 | 64 | 4 | 6 | [16, 8, 4, 2] | average pooling | 8 | 5 | 10 | | 2 | |
| SWAN | | | | | | | | | | | | |
| PSM | 192 | | | | | | | | | | | |
| SMAP | | | | | | | | | | | | |

| | **TimesNet** | | | | | **TimeMixer++** | | | | |
|---|---|---|---|---|---|---|---|---|---|---|
| Datasets | Num kernels | dModel | dMLP | Layers | Topk | dModel | dMLP | Layers | Down Sampling Layers | Down Sampling Window |
| GECCO | | | | | | | | | | |
| UCR | | | | | | 128 | 128 | | | |
| MSL | | | | | | | | | | |
| SWaT | 6 | 64 | 64 | 2 | 5 | | | 4 | 1 | 2 |
| SWAN | | | | | | 64 | 64 | | | |
| PSM | | | | | | | | | | |
| SMAP | | | | | | 128 | 128 | | | |

Internal Parameters

| Miscellaneous $\delta$ | Loss Threshold $\delta$ | RBF kernel Bandwidth | $\lambda$ | $\beta$ |
|---|---|---|---|---|
| 1E-09 | 1E-06 | [0.05, 0.1, 0.5, 1.0, 2.0] | 0.1 | 100 |

---

[6]Initial Learning Rate

## C. Visualizations

**Effect on GWNR.** To provide a comprehensive evaluation of our GWNR strategy's effectiveness, we present visualizations of the reconstruction residuals for several backbones on the PSM and SMAP datasets. For each set of plots, we analyze a randomly sampled sequence of normal data from the test set, with a length corresponding to the backbone's native input size, to validate the properties of the residuals.

The visualizations clearly demonstrate that GWNR substantially reduces the temporal correlation in the residuals compared to the vanilla MSE-trained models. This effect is formally evaluated by examining whether the autocorrelation coefficients fall within the standard 95% confidence interval for white noise (gray area). For a sequence of length $N$, this interval is defined as:

$$\left[-\frac{z_{\alpha/2}}{\sqrt{N}}, \frac{z_{\alpha/2}}{\sqrt{N}}\right].$$

In this case, the interval is $[-0.196, 0.196]$. In the ACF plots, the correlation produced by our method are markedly diminished and consistently fall within these bounds, providing strong evidence that the temporal dependencies have been successfully modeled.

Regarding Gaussianity, the Q-Q plots reveal two key improvements. First, a significant reduction in the variance of the residuals is observed across all backbones, with the overall variance typically reduced by at least an order of magnitude. Second, the residuals are better centered around zero, indicating a reduction in systematic bias, and the linearity of the points in the Q-Q plots is also visibly improved, confirming that the residual distribution more closely approximates a Gaussian distribution (red dash line).

**Anomaly Scores.** To provide further qualitative evidence, we present anomaly score sequences from various datasets and backbones. Each visualization displays a randomly sampled sequence of length 500 from the test set that includes anomalous points, allowing for a direct comparison of the anomaly scores before and after applying COGNOS. As observed in Figures 7 and 10, COGNOS effectively suppresses the erratic fluctuations and spurious spikes present in non-anomalous regions, thereby enhancing the signal of true anomalous onset. Similarly, Figure 8 illustrates that the score becomes markedly more stable, capable of sustaining a high magnitude throughout the duration of an anomaly.

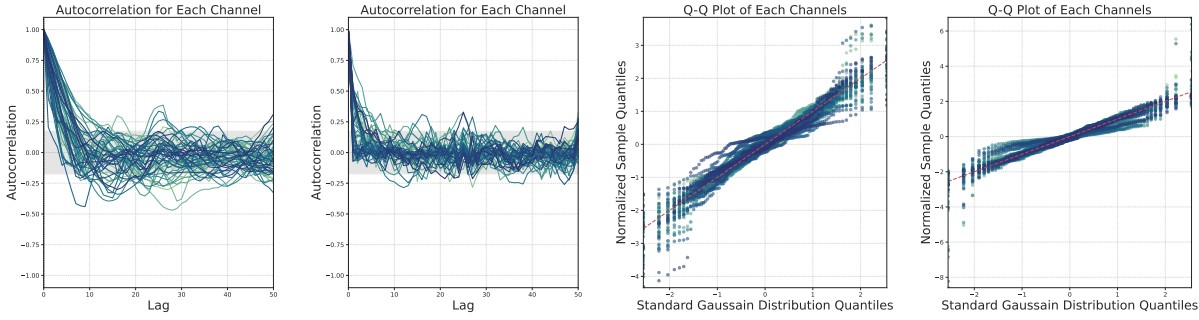

*Figure 5.* Residual analysis on the SWaT dataset using TimesNet. Left: vanilla method, Right: COGNOS.

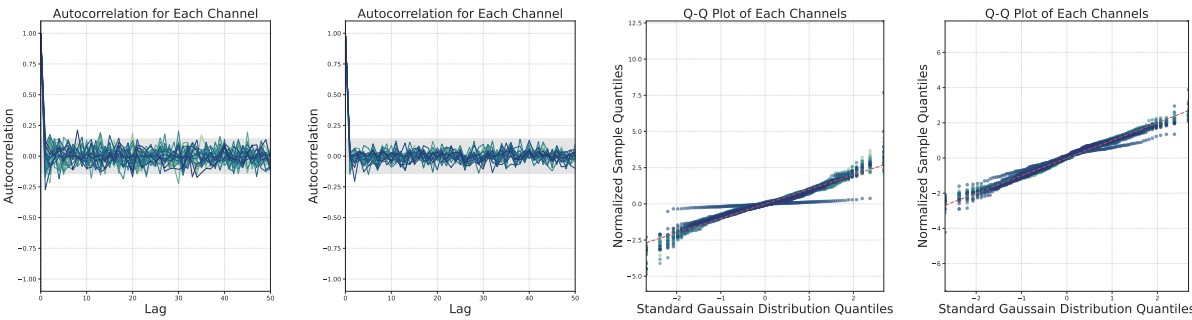

*Figure 6.* Residual analysis on the PSM dataset using TimeMixer++. Left: vanilla method, Right: COGNOS.

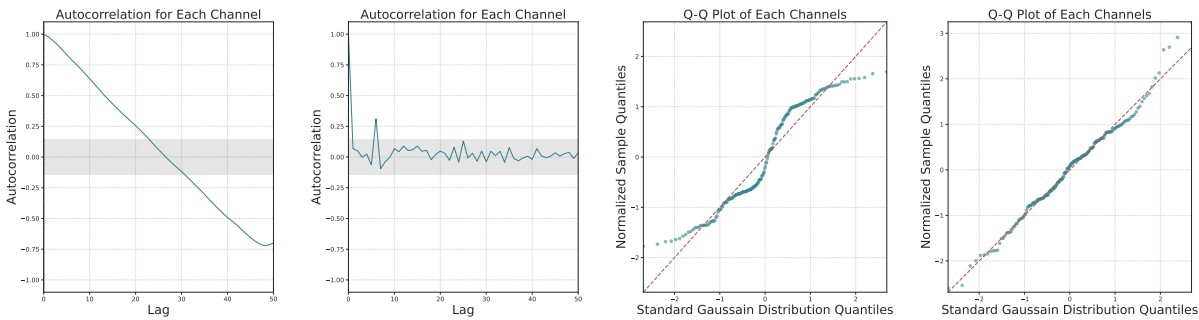

*Figure 7.* Residual analysis on the SMAP dataset using CrossAD. Left: vanilla method, Right: COGNOS.

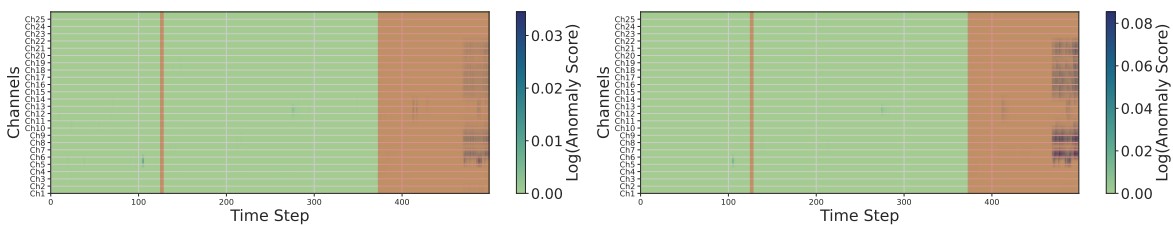

*Figure 8.* Qualitative comparison of anomaly scores on PSM using TimeMixer++. Left: vanilla method, Right: COGNOS.

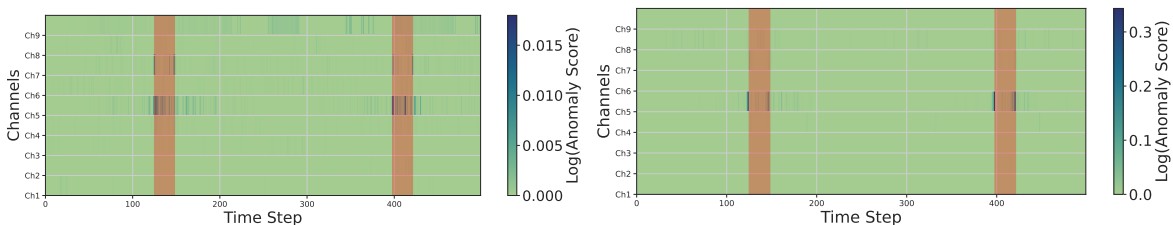

*Figure 9.* Qualitative comparison of anomaly scores on GECCO using ModernTCN. Left: vanilla method, Right: COGNOS.



*Figure 10.* Qualitative comparison of anomaly scores on PSM using KAN-AD. Left: vanilla method, Right: COGNOS.

# D. Detailed Experiments

In this study, we evaluated performance on an AMD EPYC 7002 CPU with 48 GB of RAM, complemented by an NVIDIA RTX 4090 GPU equipped with 24 GB of VRAM. The source code was developed using Python 3.10 and PyTorch 2.1.0, and executed on an Ubuntu 22.04 operating system with CUDA 11.8 support. To ensure reproducibility, we fixed the random seed at '2025' across all experiments.

## D.1. Main Results

To ensure a comprehensive and unbiased assessment, we employ a multi-dimensional metric suite covering both threshold-dependent and threshold-independent perspectives. We report the standard **Point-Adjustment F1** (Std-F1) (Xu et al., 2018) for consistency with prior literature, while mitigating its potential leniency by incorporating the **Affiliation-based F1** (Aff-F1) (Huet et al., 2022). The latter provides a theoretically grounded measure based on the temporal proximity between ground truth and predicted ranges. Furthermore, to evaluate detection robustness free from specific threshold selection, we adopt the Volume Under the Surface (VUS) framework (Paparrizos et al., 2022). Specifically, we report **Range-AUC-ROC** (R-A-R) and **Range-AUC-PR** (R-A-P) to capture range-based structural overlap, alongside **VUS-ROC** (V-R) and **VUS-PR** (V-P), which holistically measure performance stability across varying buffer sizes and temporal misalignments. For UCR datasets, we report average results across all 250 subsets. Detailed results are presented in Tables 7 and 8.

*Table 7.* Multi-metric evaluation of anomaly detection performance between the Vanilla method and COGNOS (Ours) across six datasets. Best results are highlighted in **bold**.

| Models | Datasets | GECCO | | MSL | | PSM | | SMAP | | SWAN | |
|---|---|---|---|---|---|---|---|---|---|---|---|
| | Metircs | Vanilla | Ours | Vanilla | Ours | Vanilla | Ours | Vanilla | Ours | Vanilla | Ours |
| Autoformer | Std-F1 | **0.4826** | **0.7507** | 0.7724 | **0.9321** | 0.9098 | **0.9759** | 0.6847 | **0.7829** | 0.7336 | **0.7950** |
| | Aff-F1 | **0.3929** | **0.9426** | 0.4008 | **0.9388** | 0.5292 | **0.8743** | 0.6266 | **0.8521** | 0.1828 | **0.3604** |
| | R-A-R | **0.9758** | **0.9825** | 0.6942 | **0.7084** | 0.6667 | **0.6954** | 0.5662 | **0.6109** | 0.9398 | **0.9433** |
| | R-A-P | **0.5918** | 0.5471 | 0.2366 | **0.2446** | 0.4851 | **0.5235** | **0.1796** | 0.1652 | 0.9362 | **0.9382** |
| | V-R | **0.9930** | 0.9920 | 0.6684 | **0.6884** | 0.6349 | **0.6650** | 0.5552 | **0.5882** | 0.9532 | **0.9542** |
| | V-P | **0.6447** | 0.5069 | 0.1943 | **0.2039** | 0.4352 | **0.4740** | **0.1638** | 0.1490 | 0.9074 | **0.9125** |
| KANAD | Std-F1 | 0.4738 | **0.7466** | 0.8115 | **0.9165** | 0.8946 | **0.9744** | 0.6237 | **0.7085** | 0.7384 | **0.8049** |
| | Aff-F1 | 0.3785 | **0.9465** | 0.5360 | **0.9157** | 0.6223 | **0.8628** | 0.6666 | **0.8491** | 0.2066 | **0.3936** |
| | R-A-R | **0.9761** | 0.9753 | 0.6646 | **0.6806** | **0.7005** | 0.6799 | **0.6136** | 0.6023 | 0.9227 | **0.9374** |
| | R-A-P | **0.5810** | 0.5040 | 0.2030 | **0.2135** | **0.5092** | 0.4740 | **0.1757** | 0.1567 | 0.9210 | **0.9301** |
| | V-R | 0.8825 | **0.9931** | 0.6450 | **0.6596** | **0.6846** | 0.6624 | **0.5969** | 0.5831 | 0.9414 | **0.9460** |
| | V-P | 0.0993 | **0.5683** | 0.1742 | **0.1799** | **0.4724** | 0.4432 | **0.1612** | 0.1450 | 0.8950 | **0.8983** |
| TimesNet | Std-F1 | 0.7438 | **0.7444** | 0.7944 | **0.9160** | 0.9581 | **0.9751** | **0.7876** | 0.6636 | 0.7371 | **0.8313** |
| | Aff-F1 | 0.8417 | **0.9342** | 0.4868 | **0.9171** | 0.7828 | **0.8672** | 0.5289 | **0.8301** | 0.1961 | **0.5249** |
| | R-A-R | 0.9443 | **0.9709** | **0.6968** | 0.6663 | 0.6742 | **0.7105** | 0.5661 | **0.5840** | 0.9387 | **0.9413** |
| | R-A-P | 0.4226 | **0.4477** | **0.2323** | 0.2067 | 0.4932 | **0.5290** | **0.1576** | 0.1506 | 0.9366 | **0.9394** |
| | V-R | 0.9874 | **0.9901** | **0.6713** | 0.6500 | 0.6408 | **0.6805** | 0.5539 | **0.5718** | **0.9544** | 0.9514 |
| | V-P | **0.4942** | 0.4753 | **0.1925** | 0.1760 | 0.4402 | **0.4787** | **0.1456** | 0.1428 | 0.9128 | **0.9161** |

## D.2. Evaluation Protocol: Thresholding Pipeline

To ensure absolute fairness and reproducibility, we adopt a unified global quantile threshold for each dataset. The threshold percentile is selected via a constrained search within the range $[98\%, 99.5\%]$, chosen to establish a stable operating point where diverse backbone architectures yield reliable and comparable performance. This single threshold is applied uniformly to all models evaluated on a given dataset; no per-model exhaustive F1-maximization or dynamic thresholding is performed. For the Affiliated F1 (Aff-F1) metric, predicted anomaly ranges are constructed as follows: after binarizing anomaly scores using the global quantile threshold, any contiguous sequence of anomalous points is grouped into a single predicted event

*Table 8.* Comparison of anomaly detection performance between the Vanilla method and Ours (Ours) across seven datasets. The table reports the **Standard-F1** and **Affiliated-F1** metric. Best results are highlighted in **bold**.

| **Models** | | Autoformer | | CrossAD | | DLinear | | KANAD | | LSTMAE | |
|---|---|---|---|---|---|---|---|---|---|---|---|
| **Datasets** | Metircs | Vanilla | Ours | Vanilla | Ours | Vanilla | Ours | Vanilla | Ours | Vanilla | Ours |
| **GECCO** | Std-F1 | 0.4826 | **0.7507** | 0.4401 | **0.7228** | 0.4326 | **0.7400** | 0.4738 | **0.7466** | 0.4442 | **0.7619** |
| | Aff-F1 | 0.3929 | **0.9426** | 0.3115 | **0.9320** | 0.4549 | **0.9384** | 0.3785 | **0.9465** | 0.3265 | **0.9463** |
| **MSL** | Std-F1 | 0.7724 | **0.9321** | 0.7320 | **0.9315** | 0.8196 | **0.9322** | 0.8115 | **0.9165** | 0.8108 | **0.9320** |
| | Aff-F1 | 0.4008 | **0.9388** | 0.4135 | **0.9386** | 0.5654 | **0.9432** | 0.5360 | **0.9157** | 0.5501 | **0.9460** |
| **PSM** | Std-F1 | 0.9098 | **0.9759** | 0.9256 | **0.9740** | 0.9582 | **0.9747** | 0.8946 | **0.9744** | 0.9138 | **0.9759** |
| | Aff-F1 | 0.5292 | **0.8743** | 0.5904 | **0.8628** | 0.7728 | **0.8615** | 0.6223 | **0.8628** | 0.6798 | **0.8629** |
| **SMAP** | Std-F1 | 0.6847 | **0.7829** | 0.4639 | **0.6438** | 0.6426 | **0.6641** | 0.6237 | **0.7085** | **0.7012** | 0.5950 |
| | Aff-F1 | 0.6266 | **0.8521** | 0.4871 | **0.7857** | 0.6728 | **0.8437** | 0.6666 | **0.8491** | 0.6183 | **0.7770** |
| **SWAN** | Std-F1 | 0.7336 | **0.7950** | 0.7358 | **0.8354** | 0.7435 | **0.8159** | 0.7384 | **0.8049** | 0.7417 | **0.8150** |
| | Aff-F1 | 0.1828 | **0.3604** | 0.1829 | **0.5374** | 0.2268 | **0.4402** | 0.2066 | **0.3936** | 0.2193 | **0.4329** |
| **SWAT** | Std-F1 | 0.8930 | **0.9633** | 0.8621 | **0.9617** | 0.8999 | **0.9618** | 0.8558 | **0.9617** | 0.9062 | **0.9622** |
| | Aff-F1 | 0.3967 | **0.9571** | 0.2973 | **0.9511** | 0.4283 | **0.9542** | 0.2905 | **0.9543** | 0.4677 | **0.9521** |
| **UCR** | Std-F1 | 0.2827 | **0.2879** | 0.2092 | **0.3167** | **0.2738** | 0.2583 | **0.2558** | 0.2358 | 0.2386 | **0.2411** |
| | Aff-F1 | 0.4702 | **0.7427** | 0.3383 | **0.7412** | 0.4782 | **0.7325** | 0.4409 | **0.7279** | 0.4074 | **0.7328** |

| **Models** | | MICN | | ModernTCN | | TimeMixer++ | | TimesNet | |
|---|---|---|---|---|---|---|---|---|---|
| **Datasets** | Metircs | Vanilla | Ours | Vanilla | Ours | Vanilla | Ours | Vanilla | Ours |
| **GECCO** | Std-F1 | **0.7772** | 0.7423 | 0.6086 | **0.7335** | **0.7611** | 0.7302 | 0.7438 | **0.7444** |
| | Aff-F1 | 0.9141 | **0.9406** | 0.5646 | **0.9331** | 0.7936 | **0.9356** | 0.8417 | **0.9342** |
| **MSL** | Std-F1 | 0.7979 | **0.9161** | 0.6097 | **0.9160** | 0.8070 | **0.9430** | 0.7944 | **0.9160** |
| | Aff-F1 | 0.4898 | **0.9150** | 0.3545 | **0.9177** | 0.5179 | **0.9450** | 0.4868 | **0.9171** |
| **PSM** | Std-F1 | 0.9422 | **0.9756** | 0.9155 | **0.9752** | 0.9491 | **0.9744** | 0.9581 | **0.9751** |
| | Aff-F1 | 0.7545 | **0.8708** | 0.6903 | **0.8724** | 0.7018 | **0.8661** | 0.7828 | **0.8672** |
| **SMAP** | Std-F1 | **0.8151** | 0.6314 | **0.8694** | 0.7229 | **0.4924** | 0.2206 | **0.7876** | 0.6636 |
| | Aff-F1 | 0.6915 | **0.7856** | 0.7321 | **0.8262** | 0.3764 | **0.4063** | 0.5289 | **0.8301** |
| **SWAN** | Std-F1 | 0.7374 | **0.8098** | 0.7325 | **0.8155** | 0.7377 | **0.8249** | 0.7371 | **0.8313** |
| | Aff-F1 | 0.2085 | **0.4307** | 0.1850 | **0.4614** | 0.1889 | **0.5374** | 0.1961 | **0.5249** |
| **SWAT** | Std-F1 | 0.9503 | **0.9637** | 0.8238 | **0.9640** | 0.8539 | **0.9615** | 0.9398 | **0.9644** |
| | Aff-F1 | 0.6111 | **0.9575** | 0.2495 | **0.9643** | 0.2946 | **0.9509** | 0.5873 | **0.9601** |
| **UCR** | Std-F1 | **0.3661** | 0.3248 | **0.2906** | 0.2783 | **0.3564** | 0.3057 | 0.3531 | **0.3626** |
| | Aff-F1 | 0.6402 | **0.7540** | 0.5369 | **0.7452** | 0.6232 | **0.7390** | 0.6505 | **0.7619** |

range. The ARKS offline calibration computes system matrices strictly from the normal training set and has no access to test labels, ensuring zero information leakage.

## D.3. Ablation Study

We present a comprehensive ablation study to dissect the contributions of COGNOS's key components. This includes evaluations of the full COGNOS framework, comparisons with alternative filtering techniques (i.e., GWNR combined with moving average (MA) and low-pass (LP) filters) [7] , substitution of GWNR with standard MSE training (w/o GWNR + ARKS), apply heuristic filtering with standard MSE training (w/o GWNR + w/ Filter), removal of the filtering step while directly using MSE as the anomaly score (w/ GWNR + w/o Filter), and the vanilla baseline employing MSE for both training and anomaly scoring (w/o GWNR + w/o Filter). As evidenced in Tables 9, COGNOS consistently achieves superior and stable performance across diverse models and datasets.

*Table 9.* Ablation Study of COGNOS Components and Filtering Methods. The best results are highlighted in **bold**, and the second-best are underlined.

| Datasets | | | | GECCO | | MSL | | PSM | | SMAP | | SWAN | |
|---|---|---|---|---|---|---|---|---|---|---|---|---|---|
| GWNR | ARKS | MA | LP | Std-F1 | Aff-F1 | Std-F1 | Aff-F1 | Std-F1 | Aff-F1 | Std-F1 | Aff-F1 | Std-F1 | Aff-F1 |
| ✗ | | | | 0.4738 | 0.3785 | 0.8115 | 0.5360 | 0.8946 | 0.6223 | 0.6237 | 0.6666 | 0.7384 | 0.2066 |
| ✗ | ✓ | | | 0.4740 | 0.4845 | 0.8083 | 0.5285 | 0.9433 | 0.6689 | 0.6035 | 0.6024 | 0.7403 | 0.2152 |
| ✗ | | ✓ | | 0.3886 | 0.4259 | 0.1601 | 0.2530 | 0.8546 | 0.3179 | 0.3529 | 0.3040 | 0.7321 | 0.1792 |
| ✗ | | | ✓ | 0.3985 | 0.4292 | 0.1951 | 0.2650 | 0.8566 | 0.4108 | 0.4372 | 0.4340 | 0.7321 | 0.1778 |
| ✓ | | | | 0.3440 | 0.4098 | 0.8704 | 0.5917 | 0.9160 | 0.4910 | 0.6027 | 0.2980 | 0.7348 | 0.1872 |
| ✓ | | ✓ | | 0.3531 | 0.4123 | 0.1626 | 0.2564 | 0.7979 | 0.1845 | 0.3646 | 0.2125 | 0.7321 | 0.1792 |
| ✓ | | | ✓ | 0.3539 | 0.4236 | 0.1662 | 0.2584 | 0.8006 | 0.2302 | 0.3658 | 0.2081 | 0.7321 | 0.1776 |
| ✓ | ✓ | | | **0.7466** | **0.9465** | **0.9165** | **0.9157** | **0.9744** | **0.8628** | **0.7085** | **0.8491** | **0.8049** | **0.3936** |

## D.4. Computational Efficiency

*Table 10.* Computational Efficiency Analysis. We report the **average execution time (ms) per iteration** (Training) and **per sample point** (Inference). The value following "+" denotes the additional overhead introduced by our COGNOS framework. Note that the training overhead varies due to the data-dependent dynamic masking mechanism. We set batch size= 128 and sequence length = 128

| Datasets | Models | Autoformer | | KANAD | | TimesNet | |
|---|---|---|---|---|---|---|---|
| | | Vanilla | COGNOS | Vanilla | COGNOS | Vanilla | COGNOS |
| **GECCO** | Train | 62.53 | +41.648 | 39.94 | +48.4 | 101.34 | +171.49 |
| | Inference | 0.1782 | +0.1274 | 0.0847 | +0.1467 | 0.1694 | +0.1513 |
| **MSL** | Train | 65.32 | +210.196 | 146.68 | +354.92 | 97.57 | +308.209 |
| | Inference | 0.1829 | +0.0831 | 0.2660 | +0.144 | 0.1463 | +0.1425 |
| **PSM** | Train | 53.30 | +82.662 | 97.31 | +206.687 | 103.89 | +257.55 |
| | Inference | 0.2150 | +0.1031 | 0.2459 | +0.1387 | 0.2054 | +0.1327 |
| **SMAP** | Train | 96.23 | +41.721 | 19.39 | +28.792 | 145.31 | +183.52 |
| | Inference | 0.2069 | +0.1055 | 0.0391 | +0.1396 | 0.2068 | +0.145 |
| **SWAN** | Train | 87.64 | +164.116 | 216.54 | +355.01 | 153.57 | +359.27 |
| | Inference | 0.2482 | +0.0838 | 0.3546 | +0.1268 | 0.2190 | +0.1402 |
| **AVG** | Train | 73.01 | +108.0686 | 103.97 | +198.7618 | 120.34 | +256.0078 |
| | Inference | 0.2062 | +0.1006 | 0.1981 | +0.1392 | 0.1894 | +0.1423 |

**Inference Overhead.** The ARKS filter executes serially due to its temporal recurrence, introducing approximately **0.13 ms/point** of additional latency relative to vanilla inference. However, this comparison is partially misleading: the

---

[7]To ensuring a fair comparison, we selected parameters that empirically offered the best trade-off between noise suppression and signal preservation for the datasets used: For Moving Average, we utilized a sliding window size of $w = 20$, for Low-Pass Filter, we employed a Butterworth low-pass filter with a normalized cutoff frequency of $f_c = 0.05$ (Nyquist frequency scale).

reported vanilla numbers are measured in a parallelized offline evaluation setting, whereas in real-world industrial streaming deployments, data arrives sequentially and both vanilla and COGNOS methods are restricted to serial processing. Under realistic streaming conditions, the effective latency gap narrows substantially.

**Training Overhead.** The GWNR loss increases per-iteration training time by an average of 187 ms, primarily driven by the frequency-domain spectral flatness computation and the multi-scale RBF kernel MMD. Crucially, this cost is strictly an **offline, one-time overhead**: it is incurred only during training and does not affect inference latency. Furthermore, as shown in Table 4, the total training duration with COGNOS often remains within the natural variability between different backbone architectures, rendering it a practical overhead given the substantial detection performance gains.

### D.5. Comparison with Anomaly Transformer

To provide a broader empirical context, we evaluate the standalone Anomaly Transformer (Xu et al., 2022) under our unified evaluation protocol (see Appendix D.2). Note that Anomaly Transformer is not enhanced by COGNOS, as it optimizes Association Discrepancy rather than MSE reconstruction residuals and therefore falls outside COGNOS's applicable scope (Section 6). Results are reported in Table 11.

*Table 11.* Anomaly Transformer performance under the unified quantile threshold protocol across five representative datasets.

| Dataset | Precision | Recall | Std-F1 | Aff-P | Aff-R | Aff-F1 |
|---------|-----------|--------|--------|-------|-------|--------|
| PSM | 0.9739 | 0.9550 | 0.9644 | 0.5429 | 0.7366 | 0.6251 |
| MSL | 0.9189 | 0.9515 | 0.9349 | 0.5187 | 0.9648 | 0.6746 |
| SWaT | 0.9024 | 0.9982 | 0.9479 | 0.5576 | 0.9760 | 0.7097 |
| SMAP | 0.9247 | 0.9745 | 0.9489 | 0.5125 | 0.9814 | 0.6733 |
| GECCO | 0.3737 | 0.4781 | 0.4195 | 0.5630 | 0.8612 | 0.6809 |

