# OpenReview forum: "COGNOS: Universal Enhancement for Time Series Anomaly Detection via Constrained Gaussian-Noise Optimization and Smoothing"
_ICML.cc/2026/Conference — ICML 2026 regular_

### Official Review · Reviewer_LkND · 2026-03-09

**Soundness:** 3
**Presentation:** 2
**Significance:** 3
**Originality:** 3
**Overall Recommendation:** 4
**Confidence:** 3

**Summary:**

This paper proposes COGNOS, a model-agnostic enhancement framework for reconstruction-based time series anomaly detection (TSAD). The paper starts from the observation that reconstruction residuals produced by models trained with standard MSE objectives are often temporally correlated and non-Gaussian, which may lead to unstable anomaly scores. To address this, the authors introduce two coupled components: (1) Gaussian-White-Noise Regularized (GWNR) Loss, which aims to shape residuals toward Gaussian white noise during training through spectral and distributional constraints, and (2) Adaptive Residual Kalman Smoother (ARKS), which uses offline calibration and an adaptive Kalman-style filtering mechanism to denoise anomaly scores at inference time. The method is evaluated on multiple TSAD datasets and several backbone models, and the paper reports substantial gains, especially in event-based metrics such as Aff-F1.
Overall, the paper's central domain concerns reconstruction-based time series anomaly detection and the statistical reliability of residual-based anomaly scoring. Overall, its core contribution concerns augmenting existing TSAD backbones by explicitly regularizing residual statistics and applying an adaptive filtering mechanism at inference time.

**Compliance With Llm Reviewing Policy:**

Affirmed.

**Final Justification:**

The authors’ rebuttal has adequately addressed my main concerns. Accordingly, I am raising my score to 4.

**Key Questions For Authors:**

1. The experiments compare the proposed method with several reconstruction-based TSAD backbones. However, widely used methods such as Anomaly Transformer or TranAD are not included. Because these are commonly used baselines in TSAD research, could the authors clarify whether there was a specific reason for not including them in the comparisons?

2. In the appendix, the main results tables present detailed scores for each dataset. It might be helpful to include additional summary statistics, such as the average score across datasets or the number of wins against baselines, to make the overall effectiveness of the proposed method easier to interpret.

3. Table 10 reports the computational cost of the method, but there is limited discussion of this table in the text. From the reported numbers, the training and inference time appear to increase noticeably relative to the baselines. Could the authors provide additional discussion on this trade-off between performance gains and computational overhead?

4. In the ablation study, the paper evaluates some design choices, but it is unclear whether results are available for removing each major component individually (e.g., removing GWNR or removing ARKS). Such results would help clarify the relative contribution of each component to the overall performance.

**Limitations:**

While the proposed framework shows promising empirical improvements, several limitations remain. First, the method introduces additional components, including the residual regularization objective and the ARKS filtering mechanism, which increase the overall complexity of the pipeline compared to standard reconstruction-based TSAD methods. In particular, the additional spectral constraints, distribution matching terms, and calibration procedures may lead to increased training and inference costs. Although the paper reports computational statistics, the trade-off between detection performance and computational efficiency could be further analyzed.
Second, the framework is designed under the assumption that encouraging residuals to approximate Gaussian white noise leads to more reliable anomaly scores. While this assumption is intuitively appealing, it may not always hold in real-world time series where residual distributions can be affected by nonstationarity, seasonality, or structured noise patterns.
Finally, although the paper evaluates the method across several datasets and backbones, additional comparisons with recent TSAD models and stronger post-processing baselines would further strengthen the empirical validation of the approach.

**Strengths And Weaknesses:**

Strengths
1. A practical strength is that the method is positioned as a model-agnostic enhancement rather than a new backbone. The paper evaluates the method across a relatively broad set of architectures, including forecasting, reconstruction, and recent TSAD models, which is useful from an applied perspective.

2. The experiments span seven datasets and multiple backbone models, and the paper reports improvements across many settings, especially on Aff-F1. The results make the empirical section more convincing than a narrow single-backbone demonstration.

3. The paper provides a clear mathematical description of the proposed framework and carefully formulates each component, including the residual regularization objective and the adaptive filtering mechanism. This formulation helps the reader understand the intended statistical behavior of the residuals and how each component contributes to the overall objective.

Weakness
1. The method combines several existing tools:
- reconstruction loss,
- spectral whitening / entropy-style constraints,
- MMD-based distribution matching,
- wavelet decomposition,
- Kalman filtering with adaptive switching.
The integration is nontrivial, but the paper does not fully clarify which part is genuinely new versus carefully engineered composition. As written, the novelty appears more in the system design and coupling than in a fundamentally new algorithmic idea.

2. The paper emphasizes that event-based metrics improve strongly, while some range-based metrics decrease in certain cases. This is acknowledged, which is good, but the discussion still tends to frame these decreases as theoretically acceptable rather than carefully analyzing when the method may actually be less suitable. In practice, some applications care about persistence and full anomaly coverage, not only onset detection. The trade-off deserves a more balanced discussion.

3. The paper provides timing numbers, which is appreciated, but the training overhead appears substantial in several cases. The total engineering and training burden seems to nontrivial.

4. The paper is readable overall, but several claims are phrased too strongly, and some wording feels more promotional than analytical. There are also places where the statistical language is used somewhat loosely.

---

> ### Author Rebuttal · Authors · 2026-03-30
>
> **We sincerely thank you for your highly constructive feedback, which highlights the practical strength, empirical comprehensiveness, and mathematical clarity of our work. To optimize space, we have consolidated our gratitude here and address your insightful concerns below.**
>
> $\text{To Q1:}$
>
> The exclusion of models like Anomaly Transformer or TranAD was **not an intentional omission**. COGNOS is specifically designed to address the statistical misalignment inherent in the widely used **MSE based training and inference paradigm**.
>
> For instance, Anomaly Transformer primarily optimizes Association Discrepancy, instead of relying purely on MSE reconstruction residuals to represent the noise term under the Wold decomposition. To clarify this, we will explicitly define the applicability boundaries of COGNOS in the "Limitations" section of the revised manuscript.
>
> $\text{To Q2:}$
>
> In the revised manuscript, we will include additional summary statistics, to make the overall effectiveness of our method easier to interpret.
>
> $\text{To Q3 W3:}$
>
> First, regarding the inference overhead: The gap observed in Tables 4 and 10 primarily stems from differences in GPU **parallelism** during offline evaluation. The vanilla method can compute multiple windows in parallel during testing, whereas ARKS must **execute serially** due to its temporal recurrence. However, in practical industrial streaming TSAD, datas arrive sequentially, meaning the vanilla method is also **restricted to serial processing**. Therefore, in a real-world deployment environment, the latency of approximately 0.13 ms/point introduced by COGNOS is an acceptable overhead.
>
> Second, regarding the training overhead: Training time does increase due to the inclusion of frequency-domain whitening and the multi-scale RBF kernel MMD computations. However, this is strictly an offline **upfront cost**. As shown in our ablation studies (Tables 3 and 9), GWNR is indispensable; it shapes the ideal Gaussian white noise conditions necessary to ensure that ARKS filtering achieves its theoretical optimum.
> Because these computations occur exclusively during the offline phase, we believe that **trading increased offline training time for significant gains in online detection performance and stability represents a highly valuable engineering trade-off.**
>
> We will include a detailed discussion of this tradeoff in the revised manuscript.
>
> $\text{To Q4:}$
>
> We have indeed performed the comprehensive component-wise ablation study you suggested, and we apologize if our formatting made these results difficult to locate. Specifically, **Table 3 in the main text and Table 9 in the Appendix** evaluate the individual contribution of each component as follows:
>
> 1. Removing both GWNR and ARKS (the Vanilla Baseline, **row 1**).
> 2. Removing GWNR but retaining ARKS (exploring the limitations of filtering with non-ideal residuals, **row 2**).
> 3. Retaining GWNR but removing ARKS, using vanilla MSE anomaly scores instead (verifying the effect of pure regularization, **row 3 in Table 3, row 5 in Table 9**).
> 4. The complete COGNOS framework (the **last row**).
>
> These results demonstrate that applying either GWNR or ARKS in isolation yields only marginal or unstable improvements; combining them is necessary to unlock their synergistic effects.
>
> $\text{To W1:}$
>
> We agree that the individual mathematical tools we employ are well established in the literature. Our fundamental novelty, however, lies in offering **a new perspective**: identifying the theoretical misalignment between the standard MSE objective and the perfect Wold Decomposition hypothesis in TSAD. Rather than proposing a fundamentally new empirical architecture, COGNOS is designed as a unified statistical framework derived from first principles to enforce this exact hypothesis.
>
> $\text{To W2:}$
>
> We sincerely accept with your critique. The observed trade-off fundamentally stems from handling non-stationary time series: normalization techniques (e.g., RevIN) are vital for suppressing false positives, but they inherently cause models to rapidly adapt to prolonged anomalies, leading to score decay.
>
> COGNOS deliberately amplifies this mechanism to prioritize extreme sensitivity to anomaly onsets while preventing sustained alarm fatigue. We fully concede that this design choice makes our method **less suitable for applications requiring strict duration tracking or full-coverage analysis**. Conversely, it is **highly optimized for safety-critical industrial settings where immediate "early warning" is the absolute priority.**
>
> Per your constructive suggestion, we will add discussion on this onset-vs-coverage trade-off and define the scenarios where our method may (and may not) be applied in "Limitations".
>
> $\text{To W4:}$
>
> We appreciate your constructive feedback regarding the tone and precision of our writing. In the revised manuscript, we will thoroughly review and temper any claims that appear overly strong or promotional.

---

> > ### Author Rebuttal · Reviewer_LkND · 2026-04-01
> >
> > Regarding Question 1, I still believe that including additional comparisons with other relevant baselines, such as Anomaly Transformer and TranAD, would further strengthen the empirical validation of the proposed method. In particular, comparisons with more recent anomaly detection methods would better demonstrate the applicability and competitiveness of your approach.

---

> > > ### Author Response · Authors · 2026-04-02
> > >
> > > **We sincerely thank the reviewer for the continuous engagement and for pushing us to further strengthen our empirical validation. We completely agree that incorporating highly recognized standalone TSAD baselines adds significant value to the comprehensiveness of our study.**
> > >
> > > To directly address your concern, we have rigorously reproduced and evaluated the Anomaly Transformer using our same evaluation protocol. Given the limited time in the rebuttal phase and our commitment to rigorous empirical validation, we have prioritized the evaluation on 5 representative datasets. The results are presented in the table below:
> > >
> > > | **Metric**      | **PSM** | **MSL** | **SWAT** | **SMAP** | **GECCO** |
> > > |-----------------|---------|---------|----------|----------|-----------|
> > > | Precision       | 0.9739  | 0.9189  | 0.9024   | 0.9247   | 0.3737    |
> > > | Recall          | 0.9550  | 0.9515  | 0.9982   | 0.9745   | 0.4781    |
> > > | **Std-F-score** | 0.9644  | 0.9349  | 0.9479   | 0.9489   | 0.4195    |
> > > | Aff-Precision   | 0.5429  | 0.5187  | 0.5576   | 0.5125   | 0.5630    |
> > > | Aff-Recall      | 0.7366  | 0.9648  | 0.9760   | 0.9814   | 0.8612    |
> > > | **Aff-F-score** | 0.6251  | 0.6746  | 0.7097   | 0.6733   | 0.6809    |
> > >
> > > We will include these additional comparisons in the revised manuscript to provide a more comprehensive benchmark.
> > >
> > > Furthermore, regarding the inclusion of recent anomaly detection methods, we would respectfully like to highlight that our original evaluation suite already comprises some of advanced and representative models published within the last year, including **CrossAD (NeurIPS 2025)**, **KAN-AD (ICML 2025)**, and **TimeMixer++ (ICLR 2025)**. We believe this extensive lineup, now further enriched by your valuable suggestion.
> > >
> > > **We deeply appreciate the time and constructive effort you have invested in reviewing our work. We hope these additional experiments fully resolve your remaining concerns and might positively encourage a reconsideration of your final score.**

---

### Official Review · Reviewer_Su1w · 2026-03-11

**Soundness:** 2
**Presentation:** 2
**Significance:** 2
**Originality:** 3
**Overall Recommendation:** 4
**Confidence:** 2

**Summary:**

COGNOS is a model-agnostic framework for improving reconstruction-based TSAD. It regularizes residuals toward Gaussian white noise during training (GWNR loss) and applies an adaptive Kalman smoother with a chi-squared circuit breaker to denoise anomaly scores at inference (ARKS). The two stages are designed to work in synergy: GWNR creates the statistical preconditions under which ARKS operates as an optimal estimator.

**Compliance With Llm Reviewing Policy:**

Affirmed.

**Final Justification:**

Overall the rebuttal addressed some of concerns and the idea is interesting.

**Key Questions For Authors:**

Q1: The Aff-F1 jumps to ~0.95 across nearly all backbones on SWaT and GECCO are strikingly uniform. How exactly is the anomaly threshold selected for threshold-dependent metrics, and is there any information leakage from using the training set as the ARKS calibration set?
Q2: TimeMixer++ on SMAP collapses from 0.492 to 0.081 Std-F1. Can you provide a concrete diagnosis of why  does GWNR degrade reconstruction, does ARKS calibrate poorly, or does the channel-independence assumption break down? A universal enhancement claim needs clear failure-mode characterization.

**Limitations:**

The paper lacks a limitations section and does not systematically diagnose its failure modes (e.g., TimeMixer++ SMAP Std-F1: 0.492 to 0.081), offering only brief generic explanations. Additionally, its core theoretical assumptions  Gaussian white noise residuals, Kalman MMSE optimality, channel-wise independence  are only partially validated via qualitative plots, with no formal statistical tests confirming they hold in practice.

**Strengths And Weaknesses:**

S1:
Well-motivated problem framing. The paper highlights an under-discussed mismatch between the MSE training objective and the statistical assumptions under which reconstruction error is most principled as an anomaly score.

S2:
Testing across nine architecturally diverse backbones and seven datasets spanning multiple domains is commendable.

W1:
The paper only measures relative improvement over each backbone's own baseline, never comparing the COGNOS-enhanced models against the strongest independent TSAD methods under a shared protocol.
W2:
The paper's theoretical framework Gaussian white noise residuals, Kalman MMSE optimality, circuit breaker dynamics, spatial independence  is never rigorously validated on real data.
W3: Failure modes and limitations are not discussed.

---

> ### Author Rebuttal · Authors · 2026-03-30
>
> **We sincerely thank you for recognizing our well-motivated problem framing and commending our extensive testing across diverse backbones and datasets. We also greatly appreciate your constructive feedback regarding baseline comparisons, theoretical validation, and failure mode analysis. We address your specific questions below:**
>
> $\text{To Q1:}$
>
> Regarding the anomaly threshold selection, to optimize space, **please see our response to Reviewer 8o9a's W1 for a detailed breakdown.** We confirm there is **no information leakage** in our approach. The ARKS offline calibration computes the system matrices strictly using the normal sequences from the training set. This calibration process has zero access to the test set or any anomaly labels.
>
> $\text{To Q2 W3:}$
>
> We sincerely thank the reviewer for prompting this failure-mode analysis, particularly regarding the TimeMixer++ results on SMAP. First, we must correct a **typo in our appendix**: the reported TimeMixer++ COGNOS Std-F1 on SMAP was misrecorded as 0.0808; the correct value is **0.2206**. We will update this in the final version.
>
> More importantly, this specific degradation serves as a illustration of a known flaw in the Standard Point Adjustment F1 (Std-F1) metric protocol. It represents a specific failure mode of using legacy metrics rather than a failure of our framework itself. As detailed in our **supplementary table**, the PA mechanism often yields artificially inflated  precision/recall for vanilla models. Because vanilla anomaly scores are frequently **contaminated by high-frequency noise and random spurious peaks** (visually evident in the baseline score sequences in Figures 3, 9, and 10), they often trigger "lucky hits." **Under PA rules, a single random noise spike falling within a ground-truth anomaly window counts the entire segment as detected**. When COGNOS successfully filters out this non-structural high-frequency noise, the model loses these spurious random hits. Consequently, if the backbone itself natively failed to capture the subtle deterministic trend of that anomaly, the noise-free COGNOS score will correctly miss it, leading to a visible drop in Std-F1 recall.
>
> This mechanism explains why we observe a collapse in Std-F1 for certain models. This failure-mode analysis highlights why **adopting Aff-F1 alongside Std-F1 is necessary** for a fair evaluation in TSAD.
>
> **supplementary Table (SMAP dataset):**
>
> | Metrics | LSTMAE |  | MICN |  | ModernTCN |  | TimeMixer++ |  | TimesNet |  |
> |---|---|---|---|---|---|---|---|---|---|---|
> | COGNOS | × | √ | × | √ | × | √ | × | √ | × | √ |
> | Precision | 0.9031  | 0.8288  | 0.9210  | 0.8444  | 0.9328  | 0.8756  | 0.8365  | 0.5809  | 0.8908  | 0.8598  |
> | Recall | **0.5731**  | **0.4641**  | **0.7310**  | **0.5042**  | **0.8141**  | **0.6155**  | **0.3489**  | **0.1362**  | **0.7058**  | **0.5403**  |
> | F-score | 0.7012  | 0.5950  | 0.8151  | 0.6314  | 0.8694  | 0.7229  | 0.4924  | 0.2206  | 0.7876  | 0.6636  |
> | Aff Precision | 0.8696  | 0.8115  | 0.9028  | 0.7978  | 0.9132  | 0.8293  | 0.8589  | 0.6369  | 0.8410  | 0.8390  |
> | Aff Recall | **0.4796**  | **0.7453**  | **0.5603**  | **0.7738**  | **0.6110**  | **0.8231**  | **0.2410**  | **0.2983**  | **0.3857**  | **0.8215**  |
> | Aff F-score | 0.6183  | 0.7770  | 0.6915  | 0.7856  | 0.7321  | 0.8262  | 0.3764  | 0.4063  | 0.5289  | 0.8301 |
>
>
> $\text{To W1:}$
>
> We agree that comparing with the SOTA is crucial. We would like to clarify that our selected baselines do include some of the strongest and most representative recent independent TSAD methods, **such as TimesNet, TimeMixer++, and very recent 2025 models like CrossAD and KAN-AD.**
>
> Our primary objective is not to chase SOTA metrics with a new standalone backbone, but rather to introduce a new perspective on TSAD and to offer a unified, model-agnostic enhancement to validate this perspective.
>
> $\text{To W2:}$
>
> We respectfully clarify that our framework is **validated using standard diagnostic methodologies widely accepted in statistical signal processing**. To assess residual properties, we utilized ACF and Q-Q plots. Crucially, our ACF plots explicitly include the 95% confidence bounds for white noise. As demonstrated across multiple architectures and datasets, the COGNOS-enhanced residual of normal sequence consistently fall within these intervals, confirming the suppression of temporal correlation. Similarly, the Q-Q plots tightly align with the theoretical Gaussian reference, validating our distributional assumptions.
>
> Furthermore, the Kalman MMSE optimality is validated empirically through our ablation study (Table 3). Replacing ARKS with heuristic Moving Average or Low-Pass filters leads to performance degradation. This demonstrates that our ARKS achieves optimal separation of signal and noise precisely because GWNR has engineered the requisite residual preconditions.

---

> > ### Author Rebuttal · Reviewer_Su1w · 2026-04-03
> >
> > The rebuttal helps, especially on the no-leakage point, but the main issues remain only partly answered. The SMAP/TimeMixer++ degradation is still substantial even after the typo correction, so this looks like a real failure mode or tradeoff. The theoretical validation is still mostly qualitative (ACF/Q-Q plots and ablations) rather than formal statistical testingWhile I increased my score, the paper would still be better if it included more state-of-the-art methods.

---

> > > ### Author Response · Authors · 2026-04-03
> > >
> > > **We sincerely thank you for the score increase and the ongoing constructive dialogue.** Your continued focus on the SMAP degradation highlights an important phenomenon. We agree that this numerical drop requires careful contextualization for future readers.
> > >
> > > In revised manuscript, we will discussing this specific case dedicatedly, which will help clarify the evaluation of our framework. Additionally, we highly value your suggestion regarding state-of-the-art methods. **Thank you again for helping us improve the clarity and depth of our work.**

---

### Official Review · Reviewer_Njfo · 2026-03-12

**Soundness:** 3
**Presentation:** 3
**Significance:** 3
**Originality:** 3
**Overall Recommendation:** 4
**Confidence:** 4

**Summary:**

The paper proposes COGNOS, a universal, model-agnostic enhancement framework for reconstruction-based time-series anomaly detection (TSAD). It starts from the observation that the standard use of MSE loss produces statistically unreliable reconstruction residuals, leading to noisy and unstable anomaly scores. To address this, COGNOS introduces a Gaussian-White Noise Regularization strategy during training that explicitly constrains residuals on normal data to follow a Gaussian white noise distribution. Building on this engineered statistical structure, the framework then applies an Adaptive Residual Kalman Smoother during inference to denoise raw anomaly scores in a statistically robust manner. Through extensive experiments on multiple benchmarks and diverse backbone models, the authors show that coupling residual regularization with adaptive smoothing consistently improves detection performance.

**Compliance With Llm Reviewing Policy:**

Affirmed.

**Final Justification:**

I updated my score to weak accept based on the discussion with the authors.

**Key Questions For Authors:**

In Table 2 I did not understand why COGNOS has different results as the model changes. Shouldnt Cognos's result remain the same?

Going through the code I have a concern that Point Adjustment strategy was used to update the final result. This is a strategy that uses ground truth to update test labels and inflates the results significantly. This should be avoided. Can the authors confirm that this strategy was not used?

What metrics are used in Table 1?

**Limitations:**

No limitations section.

**Strengths And Weaknesses:**

Strengths:

The assumptions are clear and the intuition behind the modeling approach is clearly explained.

The breadth of evaluation is impressive with several benchmarks and baselines.

Clear exposition of ablation studies and execution time analysis.

Figure 3 is especially interesting that shows the anomaly score change and reconstruction residuals which clearly shows why COGNOS works.

Weaknesses:

The assumption that

$X_t = f_\theta (X) + n_t + s_t$

This has an implicit assumption that any anomaly is independent of the dynamical process and is a noise. This is not true in many scenarios and only strictly holds true when anomaly is either synthetically added or when anomaly is very different from the dynamics. Even for sensor noise, the noise is highly dependent on the sensor values and is not an independent entity. Hence assuming independence between anomaly process and true dynamics is over -simplification of the problem.

Even the assumption that the noise is Gaussian may not always be true in practical cases. For example, the noise in a continuous glucose monitor sensor is a discontinuous function of the sensor value. Please refer to the 20-20 rule in CGM sensor accuracy. An example is that the error in the CGM sensor is 20 mg/dl max when CGM is < 180 mg/dl > 70 mg/dl but if CGM > 180 mg/dl the max error is 20%. This violates the Gaussian assumption.

In Table 2 I did not understand why COGNOS has different results as the model changes. Shouldnt Cognos's result remain the same?

Going through the code I have a concern that Point Adjustment strategy was used to update the final result. This is a strategy that uses ground truth to update test labels and inflates the results significantly. This should be avoided. Can the authors confirm that this strategy was not used?

What metrics are used in Table 1?

###### after rebuttal update

All my issues addressed except for my fundamental disagreement that anomaly may not be a shift in distribution. I think that is out of scope of the paper. Hence I update my score to 4.

---

> ### Author Rebuttal · Authors · 2026-03-30
>
> **We sincerely thank you for your constructive feedback and for recognizing the clarity of our assumptions, the breadth of our evaluations, the value of our ablation studies, and the insights on 20-20 rule in CGM sensor. We have carefully addressed your concerns below.**
>
> ${\text{To Q1 W3:}}$
>
> We clarify that COGNOS is a **unified enhancement module**, not a standalone predictive model. Consequently, its final anomaly score inherently relies on the representational capacity of the chosen backbone $f_\theta$. If a backbone struggles to model complex deterministic trends, the initial residuals will naturally contain structural leakage. This variation in residual quality directly impacts the state estimation within the ARKS module and can be visually confirmed by the differing ACF plots in Figure 3 (Bottom Row) and Figure 6. In summary, while COGNOS consistently elevates the performance ceiling of various backbones, the **inherent capacity of each specific backbone still defines the baseline from which improvements are made.**
>
> ${\text{To Q2 W4:}}$
>
> We completely agree that the Point Adjustment strategy can significantly inflate results and mask poor model performance. To directly address your concern: **we strictly did not use PA for our core metrics**. Evaluations including Aff-F1, R-A-R, R-A-P, V-R, and V-P were computed **entirely without PA**. We only reported the traditional PA F1 (Std-F1) solely to maintain backward compatibility with older baseline literature.
>
> ${\text{To Q3 W5:}}$
>
> We apologize for any confusion regarding Table 1. While these metrics are introduced in **Section** 5.1, we will expand their definitions in the revision to aid readability:
>
> * **Std-F1**: Evaluates a segment as a True Positive if at least one anomaly is correctly detected within the ground truth window.
> * **Aff-F1**: Calculates precision and recall based on the exact temporal proximity and intersection distance between the predicted anomaly events and the ground truth events.
> * **R-A-R and R-A-P**: Extend traditional AUC calculations to continuous segments, measuring the structural overlap between predicted and true anomaly ranges across all possible decision thresholds.
> * **V-R and V-P**: Compute the Range-AUC across a continuous spectrum of sliding buffer sizes to create a 3D volume, which corresponds to the averaged range-based measure.
>
> ${\text{To W1:}}$
>
> We fully agree that the physical origin of an anomaly is often highly dependent on the underlying dynamical state of the system. However, **Equation 1 does not assume the physical generation of the anomaly is independent. Rather, it formulates how an anomaly mathematically manifests within the residual space**. Our backbone model $f_\theta$ is tasked with capturing all predictable, state-dependent normal dynamics. Consequently, when a state-dependent anomaly occurs, it represents a breakdown of these predictable dynamics. In our framework, $\mathbf{s}_t$ mathematically quantifies this exact structural deviation from the nominal conditional distribution. **It is not assumed to be physically independent noise; it is a statistically significant shift away from the baseline stochastic noise $\mathbf{n}_t$.**
>
>
> ${\text{To W2:}}$
>
> We sincerely thank you for introducing the 20-20 rule in CGM; this is an insightful example that perfectly highlights the challenge of temporal heteroscedasticity in TSAD.
>
> While COGNOS effectively handles channel-wise heteroscedasticity through Instance Standardization, it faces limitations with **temporal heteroscedasticity within a single channel**. If the stochastic noise inherently scales with the system dynamics, our ARKS module becomes sub-optimal. This occurs because ARKS relies on a fixed observation noise covariance matrix $R$. Under stochastic noise variance inflation, a fixed $R$ would be overly sensitive and could potentially lead to false alarms.
>
> We explicitly acknowledge that **handling sustained temporal heteroscedasticity remains a fundamental challenge for the broader TSAD community**. Because the prevailing reconstruction paradigm relies heavily on the MSE objective, which intrinsically assumes homoscedasticity (**Appendix** A.1), dynamically adapting to state-dependent noise variances extends beyond the current scope of COGNOS.
>
> **Consequently, the primary intended scope of COGNOS encompasses cyber-physical and industrial systems** (such as SWaT and MSL). In these environments, sensor noise floors typically remain **relatively stable across different operational states**. Under these conditions, the homoscedastic assumption serves as a highly effective engineering approximation, **allowing ARKS to robustly isolate structural anomalies**. We will add a formal **Limitations section** in the revised manuscript to explicitly discuss this operational boundary.

---

> > ### Author Rebuttal · Reviewer_Njfo · 2026-03-31
> >
> > I only chose option c in rebuttal because of the assumption that anomaly is a significant shift in normal operation. Which it may not be. But think that should be a different paper. Except for that everything else is resolved and I will update my score.

---

> > > ### Author Response · Authors · 2026-04-01
> > >
> > > **We sincerely thank you for the score increase and the constructive engagement throughout the rebuttal process. Your insights regarding the fundamental assumptions of anomaly detection are both profound and highly practical.**
> > >
> > > In the revised manuscript, we will incorporate these insights to provide a discussion on the underlying limitations of current assumptions. We will specifically clarify the complexities you raised regarding the non-independence of sensor noise: this addition will clarify the boundaries of our current framework. Thank you again for helping us improve the depth and practical relevance of our work.

---

### Official Review · Reviewer_8o9a · 2026-03-12

**Soundness:** 2
**Presentation:** 3
**Significance:** 3
**Originality:** 3
**Overall Recommendation:** 4
**Confidence:** 3

**Summary:**

This work studies a common failure mode in reconstruction-based time series anomaly detection. Training with MSE implicitly assumes independent and identically distributed Gaussian residuals, yet residuals in practice can be autocorrelated and non-Gaussian, leading to noisy and unstable anomaly scores. The authors propose COGNOS, a model-agnostic enhancement framework with (i) a Gaussian white noise regularized loss that shapes residuals in time, frequency, and statistical domains, and (ii) an adaptive residual Kalman smoother with offline calibration and an NIS-based circuit breaker for denoising, while enabling near zero lag tracking at anomaly onsets. Experiments across 7 benchmarks and 9 backbones report consistent improvements, especially on event-based metrics such as Aff-F1.

**Compliance With Llm Reviewing Policy:**

Affirmed.

**Final Justification:**

The authors have addressed my concerns. I maintain my positive score, as I believe it is appropriate.

**Key Questions For Authors:**

Please refer to the weaknesses.

**Limitations:**

The authors have not adequately discussed the limitations and potential negative societal impacts.

**Strengths And Weaknesses:**

## Strengths

1. The method in this manuscript is conceptually well motivated and technically grounded. GWNR explicitly regularizes residual whiteness and Gaussianity, and ARKS uses an NIS based chi square test with an interpretable confidence level.

2. This manuscript provides broad empirical evidence and component ablations. It evaluates 7 datasets and 9 backbones, and provides ablations and comparisons with alternative filters (MA/LP) to support the claimed synergy between GWNR and ARKS.

3. Figures are clearly presented, making this work easy to comprehend.

## Weaknesses

1. This work's threshold dependent evaluation protocol is underspecified. While the paper explicitly distinguishes threshold dependent versus threshold independent metrics, it does not clearly describe how anomaly scores are binarized to compute Std-F1 and Aff-F1, nor how predicted ranges are constructed for Aff-F1. Given the very large gains reported on Aff-F1, the lack of a transparent thresholding and range construction pipeline makes it difficult to assess fairness and reproducibility.

2. This work's streaming latency and buffer parameter are unclear. ARKS uses a fixed buffer aggregator and outputs at $\tau \leftarrow t-k$, but the value of $k$ and its relation to window length are not specified.

3. The results in Table 7 show that in some cases the proposed method performs worse than the vanilla version. Is there any clear trend underlying these results? The paper lacks sufficient discussion on this aspect.

---

> ### Author Rebuttal · Authors · 2026-03-30
>
> **We sincerely thank you for recognizing the conceptual motivation, technical grounding, and empirical evidence of our work. We also appreciate your positive feedback on our component ablations and the clarity of our presentation. Below, we address your specific concerns regarding the evaluation protocol, streaming latency, and metric trade-offs in detail.**
>
> ${\text{To W1:}}$
>
> Regarding your concern about the binarization of anomaly scores, we initially experimented with both dynamic methods and unified fixed-quantile thresholds. We opted against dynamic methods due to their high computational cost on large datasets and extreme sensitivity to the risk probability hyperparameter $q$. To ensure absolute fairness and reproducibility, we **adopted a unified global quantile threshold for each dataset.** We strictly **avoided dynamic per-model thresholding or per-model exhaustive F1-maximization**. All models were evaluated under the exact same quantile thresholds on a given dataset.
>
> To determine this global quantile, we performed a constrained search within a strict high-percentile prior range, specifically between 98% and 99.5%. We empirically observed that a rigidly high threshold could cause catastrophic and unstable drops in true positives for certain baseline architectures, misrepresenting their actual capabilities. By slightly relaxing the threshold within this narrow band to establish a stable operating point, we ensured diverse baselines yielded stable and referenceable performance. Because this **chosen quantile was applied uniformly across all models, COGNOS was evaluated under the same strictness without model-specific advantage.**
>
> Addressing your **question on the predicted ranges for Aff-F1, we followed the standard protocol.** After binarizing the anomaly scores using the aforementioned global quantile, any contiguous sequence of anomalous points is grouped together to form a single predicted event range. We will update the manuscript to explicitly detail this pipeline.
>
> ${\text{To W2:}}$
>
> Addressing your question regarding the buffer parameter $k$ and streaming latency, we will explicitly define this relationship in the revised Algorithm 1. In our experiments, we set **$k = w - 1$, where $w$ is the window length of the backbone**. $k$ is adjustable to control the trade-off between detection latency and numerical stability, taking values $k \in (0, w - 1]$. For latency-insensitive scenarios prioritizing maximum stability, setting $k = w - 1$ ensures that a timestamp receives the full ensemble average from all overlapping windows before the ARKS filter updates. Conversely, for strict low-latency streaming, $k$ can be reduced.
>
> Furthermore, we respectfully note that this buffering delay is not a unique bottleneck introduced by COGNOS, but rather a standard characteristic of window-based paradigms. Most existing methods inherently suffer from a similar window effect latency, as they must ingest a full sequence of length $w$ and often average the anomaly scores across the sliding window to produce a reliable window-level decision. COGNOS simply formalizes this accumulation step prior to the Kalman filtering.
>
> ${\text{To W3:}}$
>
> Regarding your observation of the results in Table 7, the performance degradation in certain range-based metrics alongside dramatic improvements in event-based metrics is directly linked to the ARKS module's dynamics.
>
> As briefly discussed in **Section 5.2**, when a long-duration anomaly occurs, the Kalman Smoother triggers its Circuit Breaker and rapidly adapts its internal state to the new structural level. Once adapted, the innovation term essentially vanishes, causing the anomaly score to quickly decay back toward zero even if the physical anomaly is still ongoing. Consequently, **COGNOS acts as a hypersensitive onset detector.** It perfectly isolates the exact moment of a structural break but naturally shrinks the predicted coverage area inside long and steady-state anomaly segments. This heavily penalizes range-based overlap metrics.
>
> We view this as a **necessary trade-off to prevent sustained alarm fatigue in real-world systems**. We agree that this trend deserves more prominence, and we will expand this discussion in the revised manuscript to clearly outline the underlying mechanics of these metric trade-offs.

---

> > ### Author Rebuttal · Reviewer_8o9a · 2026-04-03
> >
> > The authors have addressed my concerns. I maintain my positive score, as I believe it is appropriate.

---

> > > ### Author Response · Authors · 2026-04-03
> > >
> > > **We are sincerely grateful for your constructive feedback and the positive final assessment of our work.** We also appreciate your recognition of our motivation; the presentation quality; and our extensive experiments. Your insightful suggestions that have significantly improved the quality and clarity of this paper.

---

### Decision · Program_Chairs · 2026-04-30

**Decision:**

Accept (regular)

**Comment:**

Based on the reviews, I recommend accepting the paper. All reviewers provided positive ratings (score 4), highlighting the technical soundness, clear methodology, and extensive empirical evaluation. Concerns about evaluation transparency and metric design were satisfactorily addressed during the rebuttal. Some issues, such as missing comparisons to certain state-of-the-art baselines, were only partially addressed. For the limitations of the assumption in anomaly modeling, reviewers acknowledged that these concerns are somewhat beyond the scope of the current work